# A Sheaf Theoretical Approach to Uncertainty Quantification of Heterogeneous Geolocation Information

**DOI:** 10.3390/s20123418

**Published:** 2020-06-17

**Authors:** Cliff A. Joslyn, Lauren Charles, Chris DePerno, Nicholas Gould, Kathleen Nowak, Brenda Praggastis, Emilie Purvine, Michael Robinson, Jennifer Strules, Paul Whitney

**Affiliations:** 1Pacific Northwest National Laboratory, Seattle, WA 98109, USA; Brenda.Praggastis@pnnl.gov (B.P.); Emilie.Purvine@pnnl.gov (E.P.); 2Pacific Northwest National Laboratory, Richland, WA 99352, USA; lauren.charles@pnnl.gov (L.C.); kathleen.nowak@cbp.dhs.gov (K.N.); pdwhitney@gmail.com (P.W.); 3College of Natural Resources, Department of Forestry and Environmental Resources, Fisheries, Wildlife, and Conservation Biology, North Carolina State University, Raleigh, NC 27695, USA; chris_deperno@ncsu.edu (C.D.); npgould@ncsu.edu (N.G.); urbanbearstudy@ncsu.edu (J.S.); 4Department of Mathematics and Statistics, American University, Washington, DC 20016, USA; michaelr@american.edu

**Keywords:** topological sheaves, information integration, consistency radius, wildlife management, stochastic linear model, Kalman filter

## Abstract

Integration of multiple, heterogeneous sensors is a challenging problem across a range of applications. Prominent among these are multi-target tracking, where one must combine observations from different sensor types in a meaningful and efficient way to track multiple targets. Because different sensors have differing error models, we seek a theoretically justified quantification of the agreement among ensembles of sensors, both overall for a sensor collection, and also at a fine-grained level specifying pairwise and multi-way interactions among sensors. We demonstrate that the theory of mathematical sheaves provides a unified answer to this need, supporting both quantitative and qualitative data. Furthermore, the theory provides algorithms to globalize data across the network of deployed sensors, and to diagnose issues when the data do not globalize cleanly. We demonstrate and illustrate the utility of sheaf-based tracking models based on experimental data of a wild population of black bears in Asheville, North Carolina. A measurement model involving four sensors deployed among the bears and the team of scientists charged with tracking their location is deployed. This provides a sheaf-based integration model which is small enough to fully interpret, but of sufficient complexity to demonstrate the sheaf’s ability to recover a holistic picture of the locations and behaviors of both individual bears and the bear-human tracking system. A statistical approach was developed in parallel for comparison, a dynamic linear model which was estimated using a Kalman filter. This approach also recovered bear and human locations and sensor accuracies. When the observations are normalized into a common coordinate system, the structure of the dynamic linear observation model recapitulates the structure of the sheaf model, demonstrating the canonicity of the sheaf-based approach. However, when the observations are not so normalized, the sheaf model still remains valid.

## 1. Introduction

There is a growing need for a representational framework with the ability to assimilate heterogeneous data. Partial information about a single event may be delivered in many forms. For instance, the location of an animal in its environment may be described by its GPS coordinates, citizen-scientist camera pictures, audio streams triggered by proximity sensors, and even Twitter text. To recover a holistic picture of an event, one must combine these pieces in a meaningful and efficient way. Because different observations have differing error models, and may even be incompatible, a model capable of integrating heterogeneous data forms has become critically necessary to satisfy sensor resource constraints while providing adequate data to researchers.

Fundamentally, we seek a theoretically justified, detailed quantification of the agreement among ensembles of sensors. From this, one can identify collections of self-consistent sensors, and determine when or where certain combinations of sensors are likely to be in agreement. This quantification should support both quantitative data and qualitative data, and continue to be useful regardless of whether the sampling rate is adequate for a full reconstruction of the scene.

Working in conjunction with recent advances in signal processing [1,2], this article shows that mathematical *sheaves* support an effective processing framework with canonical analytic methods. We ground our methodological discussion with a field experiment, in which we quantified uncertainty for a network of sensors tracking black bears in Asheville, North Carolina. The experiment provided data exhibiting a variety of error models and data types, both quantitative (GPS fixes and radio direction bearings) and qualitative (text records).

With this experiment, we demonstrate that although the sheaf methodology requires careful modeling as a prerequisite, one obtains fine-grained analytics that are automatically tailored to the specific sensor deployment and also to the set of observations.

Because there is a minimum of hidden state to be estimated, relationships between sensors—which collections of sensors yield consistent observations—can be found by inspecting the *consistency filtration*, which is naturally determined by the model. These benefits are largely a consequence of the fact that sheaves are the *canonical* structure of their kind, so that any principled specification of the interaction between heterogeneous data sources will provably recapitulate some portion of sheaf theory [2]. In short, the theory provides an algorithmic way to globalize data, and to diagnose issues when the data do not globalize cleanly.

This paper is organized as follows.

Section 2 provides context around the history of target tracking methodologies and multi-sensor fusion, including the particular role we are advocating specifically for sheaf methods.Then in Section 3 we describe the experimental setup for our data collection effort regarding tracking black bears in Asheville, North Carolina. This involved four sensors deployed among the bears, several “dummy” bear collars deployed at fixed locations in the field, and team of scientists charged with tracking their location. This experiment was designed specifically to identify a real tracking task where multi-sensor integration is required, but where the system complexity is not so large as to overwhelm the methodological development and demonstration, while still being sufficiently complexity to demonstrate the value of sheaf-based methods.Section 4 continues with a detailed mathematical development of the sheaf-based tracking model. This includes several features novel to information fusion models, including the explicit initial attention to the complexity of sensor interaction in the core sheaf models, but then the ability to equip these models with uncertainty tolerance in the form of *approximate sections*, and then to measure that both globally via a *consistency radius* and also at a more fine-grained level in terms of specific sensor dependencies in terms of a *consistency filtration*.Section 5 shows several aspects of our models interpreted through the measured data. First, overall results in terms of bear and dummy collars is provided; then results for a particular bear collar are examined in detail; and finally results for a particular time point in the measured data set is shown to illustrate the consistency filtration in particular.While our sheaf models are novel, we sought to compare them to a more traditional modeling approach, introduced in Section 6. Specifically, a statistical approach was developed to process the data once registered into a common coordinate system, a dynamic linear model which was estimated using a Kalman filter. This approach also recovered bear and human locations and sensor accuracies. Comparisons in both form and results are obtained, which demonstrate the role that sheaves as *generic* integration models can play in conjunction with *specific* modeling approaches such as these: as noted, all integration models will provably recapitulate some portion of sheaf theory [2], even if they are not first registered into a common coordinate system.We conclude in Section 7 with some general observations and discussion.

## 2. History and Context

Situations such as wildlife tracking with a variety of sensors requires the coordinated solution of several well-studied problems, most prominently that of *target tracking* and *sensor fusion*. Although the vast majority of the solutions to these problems that are proposed in the literature are statistical in nature, the sample rates available from our sensors are generally insufficient to guarantee good performance. It is for this reason that we pursued more foundational methods grounded in the geometry and topology of *sheaves*. These methods in turn have roots in *topology* and *category theory*.

### 2.1. Target Tracking Methods

Tracking algorithms have a long and storied history. Several authors (most notably [3,4]) provide exhaustive taxonomies of tracker algorithms which include both probabilistic and deterministic trackers. Probabilistic methods are typically based on optimal Bayesian updates [5] or their many variations, for instance [6,7,8].

Closer to our approach, trackers based on optimal network flow tend to be extremely effective in video object tracking [9,10,11], but require high sample rates. A tracker based on network flows models targets as “flowing” along a network of detections. The network is given weights to account for either the number of targets along a given edge, or the likelihood that some target traversed that edge. (Our approach starts with a similar network of detections, but treats the weighting very differently.) Network flow trackers are optimal when target behaviors are probabilistic [12,13], but no sampling rate bounds appear to be available. There are several variations of the basic optimal network flow algorithm, such as providing local updates to the flow once solved [14], using short, connected sequences of detections as “super detections” [15], or *K*-shortest paths, sparsely enriched with target feature information [16].

### 2.2. Multi-Sensor Fusion Methods

Data fusion is the task of forming an “alliance of data originating from different sources” [17]. There are several good surveys discussing the interface between target tracking and data fusion—such as [3,4,18]—and which cover both probabilistic and deterministic methods. Various systematic experimental campaigns have also been described [19,20]. Other authors have shown that fusing detections across sensors [21,22,23,24] yields better coverage and performance.

Our idea of using a heterogenous collection of sensors, some of which are manually operated, is similar to the sensor deployment strategy discussed in [25], which describes an approach to monitoring vehicular traffic on a road network using temporary, portable sensors. Although their underlying model is linear and satisfies a conservation law, ours need not be linear. While our sheaf-based approach to uncertainty quantification is quite different from their statistical model, we compare our results with a more standard statistical model in Section 6.

Considerable effort is typically expended developing robust features, although usually the metric for selecting features is pragmatic rather than theoretical. In all cases, though, the assumption is that sample rates are sufficiently high. For situations such as our wildlife tracking problem, this basic requirement is rarely met. Furthermore, data fusion techniques that operate on quantitative data typically require that sensors be of the same type [26,27,28]. Some prerequisite spatial registration to a common coordinate system is generally required, especially if sensor types differ [29,30,31].

The lack of a common coordinate system can make all of these approaches rather brittle to changes in sensor deployment. One mitigation for this issue is to turn to a more foundational method, for example so-called “possibilistic” information theory [32,33,34]. Here one encodes sensor models as a set of propositions and rules of inference. Data then determine the value of logical variables, from which inferences about the scene can be drawn. Allowing the variables to be valued possibilistically—as opposed to probabilistically—supports a wider range of uncertainty models. Although these methods usually can support heterogeneous collections of sensors, they do so without the theoretical guarantees that one might expect from homogeneous collections of sensors. Although the workflow for possibilistic techniques is similar to ours, the key difference is that ours relies on *geometry*. Without the geometric structure, one is faced with combinatorial complexity that scales rapidly with the number of possible sensor outputs; this frustrates the direct application of logical techniques.

Recent probabilistic models also provide a point of comparison to our sheaf methods. In [35], a multi-sensor integration system is structured within a probabilistic information fusion system. While the resulting system is specific to probabilistic robotic sensors, it may be likely that casting this in the context of a sheaf-theoretical context could provide significant value for generaliztion. Alternatively, in [36] the authors consider the problem of probabilistic target identification across multiple heterogeneous sensors, each measuring potentially different characteristics of the targets being tracked. This paper takes an evidence theory approach combining data with different levels of confidence based on foundations of Dempster-Shafer theory together with a Rayleigh distribution. While these authors consider the specific question of identifying unique targets across sensors our framework provides the ability to track various aspects of targets and the agreement (or disagreement) across sensors. It could be interesting to consider a hybrid approach leveraging their confidence aggregation methods within our sheaf-based method.

### 2.3. Sheaf Geometry for Fusion

The typical target tracking and data fusion methods tend to defer modeling until after the observations are present. If one were to reverse this workflow, requiring careful modeling before any observations are considered, then looser sampling requirements arise. This is supported by the workflow we propose, which uses interlocking local models of consistency among the observations. These interlocking local models are canonically and conveniently formalized by *sheaves*. As we discuss in Section 4, a *sheaf* is a precise specification of which sets of local data can be fused into a consistent, more global, datum [37]. While the particular kind of sheaves we exploit in this article have been discussed sporadically in the mathematics literature [38,39,40,41,42,43], our recent work cements sheaf theory to practical application.

Sheaves are an effective organization tool for heterogeneous sensor deployments [44]. Since sheaves require modeling as a *prerequisite* before any analysis occurs, encoding sensor deployments as a sheaf can be an obstacle. Many sheaf encodings of standard models (such as dynamical systems, differential equations, and Bayesian networks) have been catalogued [45]. Furthermore, these techniques can be easily applied in several different settings, for instance in air traffic control [46,47] and in formal semantic techniques [48]. Sheaf-based techniques for fundamental tasks in signal processing have also been developed [1].

The most basic data fusion question addressed by a sheaf is whether a complete set of observations constitutes a *global section*, which is a completely consistent, unified state [49]. The interface between the geometry of a sheaf and a set of observations can be quantified by the *consistency radius* [2]. Practical algorithms for data fusion arise simply as minimization algorithms for the consistency radius, for instance [50]. In this article, as in the more theoretical treatment [51], we show that sheaves provide additional, finer-grained analysis of consistency through the *consistency filtration*.

The potential significance of sheaves has been recognized for some time in the formal modeling community [37,52,53] in part because they bridge between two competing foundations for mathematics—logic and categories [54]. The study of categories is closely allied with formal modeling of data processing systems [55,56,57].

Much of the sheaf literature focuses on studying models in the absence of observations. (This is rather different from our approach, which exploits observations.) For sheaf models that have enough structure, *cohomology* is a technical tool for explaining how local observations can or cannot be fused. This has applications in several different disciplines, such as network structure [42,58,59] and quantum information [60,61]. Computation of cohomology is straightforward [1], and efficient algorithms are now available [62].

## 3. Tracking Experiment

Researchers worked collaboratively to adapt existing practices around bear management to an experimental setup designed specifically to capture the properties of the targeted problems in a tracking task where multi-sensor integration in required. The data gathered in this way directly support the initial sheaf models developed in Section 4 with an optimal level of complexity and heterogeneity. In particular, what was sought is the inclusion of two targets (a bear or dummy collar, and then the wildlife tracker) jointly engaged in a collection of sensors of heterogeneous types (including GPS, radiocollars, and hand-written reports).

### 3.1. Black Bear Study Capture and Monitoring Methodology

The black bear observational study used in this paper is located in western North Carolina and centered on the urban/suburban area in and around the city of Asheville, North Carolina. Asheville is a medium-sized city (117 km${}^{2}$) with approximately 83,000 people, located in Buncombe County in the southern Appalachian Mountain range ([63], Figure 1).

Western North Carolina is characterized by variable mountainous topography (500–1800 m elevation), mild winters, cool summers, and high annual precipitation (130–200 cm/year), mostly in the form of rainfall. Black bears occur throughout the Appalachian Mountains. The major forest types include mixed deciduous hardwoods with scattered pine [63], and pine-hardwood mix [64].

The Asheville City Boundary is roughly divided into four quadrants separated by two four-lane highways, Interstate 40, which runs east to west and Interstate 26, which runs north to south. Interstate 240 is a 9.1-mile (14.6 km) long Interstate Highway loop that serves as an urban connector for Asheville and runs in a semi-circle around the north of the city’s downtown district. We define black bear capture sites and associated locational data as ‘urban/suburban’ if the locations fall within the Asheville city limit boundary and locations outside the city boundary are considered rural.

We used landowner reports of black bears on their property to target amenable landowners to establish trap lines on or near their property, while attempting to maintain a spatially balanced sample across the city of Asheville. The sampling area included suitable plots within, or adjacent to, the city limits. We set 10-14 culvert traps in selected locations that had documented bears near their properties. We checked culvert traps twice daily, once in the morning between 0800 – 1100 and again between 1830 – 2130.

We baited traps with day old pastries. Once captured, we immobilized bears with a mixture of ketamine hydrochloride (4.0 cc at 100 mg/mL), xylazine hydrochloride (1.0 cc at 100 mg/mL) and telazol (5 cc at 100 mg/mL) at a dose of 1cc per 100 lbs. We placed uniquely numbered eartags in both ears, applied a tattoo to the inside of the upper lip, removed an upper first premolar for age determination from all bears older than 12 months [65], and inserted a Passive Integrated Transponder tag (PIT tag) between the shoulder blades. Additionally, we collected blood, any ectoparasites present, and obtained hair samples. We recorded weight, sex, reproductive status (e.g., evidence of lactation, estrus, or descended testicles), morphology, date, and capture location for each bear. We fitted bears with a global positioning system (GPS) radiocollar (Vectronics, Berlin Germany) that did not exceed 2–3% of the animal’s body weight. We administered a long-lasting pain-reliever and an antibiotic, and we reversed bears within approximately 60 min of immobilization with yohimbine hydrochloride (0.15 mg/kg).

Once black bears were captured and radiocollared, we used the virtual fence application on the GPS collars to obtain locational data every 15 min for individual bears that were inside the Asheville city limits (i.e., inside the virtual fence) and every hour when bears were outside the city limits. As a double-check, we monitored bears weekly with hand-held radio telemetry to ensure that the very high frequency (VHF) signal was transmitting correctly. We attempted this schedule on all bears to determine survival, proximity to roads and residential areas, home range size, habitat use, dispersal, and location of den sites. We stratified the locations throughout the day and night and programmed the GPS collars to send an electronic mortality message via iridium satellite to North Carolina State University (NCSU) if a bear remained immobile for more than 12 h. We immediately investigated any collar that emitted a signal to determine if it was mortality or a “slipped” collar.

### 3.2. Tracking Exercise

We used data from 12 GPS collars (6 stationary “dummy” collars and 6 “active” collars deployed on wild black bears) to design an urban tracking experiment to better estimate bear location and understand sources of error. The “dummy” collars were hidden and their locations were spatially balanced across the city limits of Asheville and hung on tree branches at various private residences. The active collars were deployed on three female black bears with the majority of their annual home range (i.e., locations) located inside the city limits. Bears N068, N083, and N024 were “tracked” three, two and one time, respectively, for the experiment.

We conducted two-hour tracking session which included 16 observations (i.e., stops) on either the dummy collar or the active bear. We stopped and recorded VHF detections on collars every 5–7 min along with data on the nearest landmark/road intersection (i.e., text description), Universal Transverse Mercator (UTM) position, elevation at each stop, compass bearing on the collar, handheld GPS position error for the observation point, and approximate distance to the collar. Also, we recorded a GPS track log during the two-hour tracking session using a second handheld GPS unit located inside the vehicle; the vehicle GPS units logged continuous coordinates (approximately every 10 s) of our driving routes during the tracking session. Finally, the GPS collars simultaneously collected locations, with associated elevation and GPS “location error”, every 15 min for the duration of the two-hour tracking shift. GPS collar data include date, time, collar ID, latitude/longitude, satellite accuracy (horizontal dilution of precision (HDOP)), fix type (e.g., 2D, 3D, or val. 3D), and elevation.

## 4. Sheaf Modeling Methodology

We now introduce the fundamentals and details of our sheaf-based tracking model of our heterogeneous information integration problem. As noted above, our wildlife tracking problem is used as an instantiation of our sheaf model intended to aim at the right level of structural complexity to demonstrate the significance of the sheaf model. As such, we introduce the mathematics below in close conjunction with the example of the bear tracking problem itself, as detailed above in Section 3. For a more extensive sheaf theory introduction, the reader is directed towards [2,40].

A sheaf is a data structure used to store information over a topological space. The topological space describes the relations among the sensors while the sheaf operates on the raw input data, mapping all the sources into a common framework for comparison. In this work, the required topological space is determined by an *abstract simplicial complex* (ASC), a discrete mathematical object representing not only the available sensors as vertices (in this case, the four physical sensors involved, three on the human-vehicle tracking team, and one on the bear itself), but also their multi-way *interactions* as higher-order *faces*. Where the ASC forms the *base space* of the sheaf, the *stalks* that sit on its faces hold the recorded data. Finally, the sheaf model is completed by the specification of *restriction functions* which model interactions of the data among sensor combinations. In this context, we can then identify *assignments* as recorded readings; *sections* as assignments which are all consistent according to the sheaf model; and *partial* assignments and sections correspondingly over some partial collection of sensors.

We then introduce some techniques which are novel for sheaf modeling. Specifically, where global or partial sections indicate data which are *completely* consistent, we introduce *consistency structures* to represent data which are only somewhat consistent. While defined completely generally, consistency structures instantiated to the bear model in particular take the form of *n*-way standard deviations. Consistency structures in turn allow us to introduce *approximate sections* which can measure the degree of consistency among sensors. The *consistency radius* (more fully developed by Robinson [2] in later work) provides a native global measure of the uncertainty among the sensors present in any reading. Beyond that, the *consistency filtration* provides a detailed breakdown of the contributions of particular sensors and sensor combinations to that overall uncertainty.

### 4.1. Simplicial Sheaf Models

First we define the concept of an abstract simplicial complex, the type of topological space used to model our sensor network.

**Definition** **1.***An* 
***abstract simplicial complex** 
*
*(ASC) over a finite base set U is a collection *Δ* of subsets of U, for which δ∈Δ implies that every subset of δ is also in *Δ*. We call each δ∈Δ with d+1 elements a* 
***d*-face**
*of *Δ*, referring to the number d as its*
* 
**dimension***
*. Zero dimensional faces (singleton subsets of U) are called*
* 
**vertices***
*, and one dimensional faces are called*
* 
**edges***
*. For γ,δ∈Δ, we say that γ is a* 
***face** 
*
*of δ (written γ⇝δ) whenever γ is a proper subset of δ.*


**Remark** **1.**
*An abstract simplicial complex *Δ* over a base set U with |U|=n can be represented in $R^{n}$ by mapping $u_{i}$ to $e_{i}$ and δ∈Δ to the convex hull of its points.*


An abstract simplicial complex can be used to represent the connections within a sensor network as follows. Take the base set *U* to be the collection of sensors for the network and take Δ to include every collection of sensors that measure the same quantity. For example, as described in Section 3 above, there are four sensors used in the tracking experiment:The *GPS* reading on the Bear Collar, denoted *G*;The *Radio VHF Device* receiver, denoted *R*;The Text report, denoted *T*; andThe *Vehicle GPS*, denoted *V*.

As shown in Table 1, the bear collar GPS and Radio VHF Device give the location of the bear, and the text report, vehicle GPS, and Radio VHF Device give the location of the researcher. We can then let U={V,R,T,G} be our base sensor set. Then our ASC Δ, representing the tracking sensor network, contains the face denoted H={V,R,T} as the set of sensors reading off in human position; the face denoted B={R,G} as those reading off on bear position; and all their subsets, so nine total faces:$$\begin{array}{ccc} \Delta = & \{ & \{V,R,T\},\\ & & \{V,T\},\{R,T\},\{V,R\},\{R,G\},\\ & & \left\{V\},\{R\},\{T\},\{G\}\;\right\} \end{array}$$

Denoting the remaining pairwise sensor interaction faces as X={V,T},Y={V,R}, and Z={R,T}, then the ASC Δ can be shown graphically on the left side of Figure 2. Here the highest dimensional faces (the human and bear positions *H* and *B* respectively) are shown, with all the sub-faces labeled. The sensors are the singleton faces (rows), and are shown in black; and the higher dimensional faces (columns) in red.

Note the presence of a solid triangle to indicate the three-way interaction *H*; the presence of the four two-way interactions as edges; and finally the presence of each sensor individually.

As a general matter, the ASC implies a topological space representing all of these multi-way interactions. Each *d*-dimensional face is a *d*-dimensional hyper-tetrahedron, which are then “glued” together or “attached” according to the configuration of the sensor interactions. The ASC shown in this example is rather simple, consisting of the “topmost” interactions as a single 2-face (the triangle of the “human” measurements) and a 1-face (the RG edge for the bear), and then the additional seven sub-faces. This simple structure has a single connected component, and no “open loops”. However, depending on the number of observables informed by a particular sensor, and their configuration, this structure can become arbitrarily complicated, with high order faces and complex connections including open loops or voids also of high dimension. The sheaf theoretical approach can represent all these interactions automatically. While such a “homological analysis” of the base space of our sheaf will not be the subject of this paper, it is great interest in computational topology generally [2,66,67,68].

The right side of Figure 2 shows the **attachment diagram** corresponding to the ASC. This is a directed acyclic graph, where nodes are faces of the ASC, connected by a directed edge pointing up from a face to its attached face (co-face) of higher dimension.

Next, given a simplicial complex, a sheaf is an assignment of data to each face that is compatible with the inclusion of faces.

**Definition** **2.** *A* 
***sheaf*S*of sets*** 
*on an abstract simplicial complex *Δ* consists of the assignment of*

*a set S(δ) to each face δ of *Δ* (called the* 
***stalk** 
*
*at δ), and*
*a function S(γ⇝δ):S(γ)→S(δ) (called the* 
***restriction map** 
*
*from γ to δ) to each inclusion of faces γ⇝δ, which obeys*
S(δ⇝λ)∘S(γ⇝δ)=S(γ⇝λ)wheneverγ⇝δ⇝λ.

*In a similar way, a 
**sheaf of vector spaces** 
assigns a vector space to each face and a linear map to each attachment. In addition, a 
**sheaf of groups** 
assigns a group to each face and a group homomorphism to each attachment.*


Intuitively, the stalk over a face is the space where the data associated with that face lives; and the restriction functions establish the grounds by which interacting data can be said to be consistent or not.

Returning to the sensors of our tracking network, a bear’s GPS collar provides its position and elevation given in units of (UTM N, UTM E, m). Thus, the stalk over the G=Bear Collar GPS vertex is $R^{3}$. Then when the researcher goes out to locate a bear he supplies two sets of coordinates and a text description of his location. The first coordinates, GPS position and elevation given in units of (lat, long, ft), come from the GPS in the vehicle the researcher is driving. Then when the researcher stops to make a measurement, he or she supplies a text description of his location. Additionally, the researcher records his or her position and elevation in (UTM N, UTM E, m), and the bear’s position (in relation to his own) off a handheld VHF tracking receiver that connects to the bear’s radiocollar, yielding a data point in $R^{5}$. The bear’s relative position is measured in polar coordinates (r,θ). The data sources and stalks are summarized in Table 2, and the left side of Figure 3 shows our ASC now adorned with the stalks on the sensor faces.

Now that we have specified the stalks for the vertices, we must decide the stalks for the higher order faces along with the restriction maps. Please note that the triangle face of the tracking ASC is formed by sensors each measuring the researcher’s location. However, the data types do not agree. Thus, to compare these measurements the information should first be transformed into common units along the edges and then passed through to the triangular face. Since multiple sensors read out in UTM for position and meters for elevation, we choose these as the common coordinate system. Then the stalk for the higher order faces of the triangle is $R^{3}$. To convert the text descriptions we use Google Maps API [69] and to convert the (lat, long, ft) readings, coming from the vehicle, we use the Python open source package utm 0.4.1 [70]. The data from the handheld tracker is already in the chosen coordinates so the restriction map to the edges of the triangle is simply a projection.

Likewise, the edge coming off the triangle connects the two sensors that report on the bear’s location. To compare the sensors we need to map them into common coordinates through the restriction maps. The sensor on the bear’s collar provides a position and elevation for the bear, but the radio VHF device only reports the bear’s position relative to the researcher. Therefore, to compare the two readings along the adjoining edge, we must forgo a degree of accuracy and use $R^{2}$ as the stalk over the edge. For the bear collar sensor this means that the restriction map is a restriction which drops the elevation reading. For the VHF data, we first convert the polar coordinates into rectangular as follows:


(1)$$\phi :R^{5}\to R^{5},\;\phi \left({\left(x,y,z,r,\theta \right)}^{T}\right)={\left(x,y,z,r\cos\left(\theta \right),r\sin\left(\theta \right)\right)}^{T}.$$


Then to obtain the position of the bear, we add the relative bear position to the human’s position.

The right side of Figure 3 thus shows the full sheaf model on the attachment diagram of the ASC. Note the stalks on each sensor vertex, text for the text reader and numerical vectors for all the others. Restriction functions are then labels on the edges connecting lower dimensional faces to their attached higher dimensional faces. UTM conversion and the Google Maps interface are on the edges coming from *V* and *T* respectively. Since the pairwise relations all share UTM coordinates, only id mappings are needed among them up to the three-way *H* face. Labels of the form $pr_{x-y}$ are projections of the corresponding coordinates (also representable as binary matrices of the appropriate form). Finally, the restriction from *R* up to *B* is the composition of the polar conversion of the final two components with the projection on the first two to predict the bear position from the radiocollar GPS, bearing, and range.

The sheaf model informs about the agreement of sensors through time as follows. At a given time *t*, each vertex is assigned the reading last received from its corresponding sensor, a data point from its stalk space. These readings are then passed through the restriction maps to the higher order faces for comparison. If the two measurements are received by an edge agree, this single value is assigned to that edge and the algorithm continues. If it is possible to assign a single value H=⟨x,y,z〉 to the {Text,Vehicle GPS,Radio VHF Device} face then all three measurements of the human’s location agree. Similarly, obtaining a single value B=⟨x,y〉 for the {Radio VHF Device,Bear Collar} edge signifies agreement among the two sensors tracking the bear. If a full assignment can be made, we call it a **global section**. Non-agreement is definitely possible, giving rise to the notion of an **assignment**.

**Definition** **3.***Let S be a sheaf on an abstract simplicial complex *Δ*. An* 
**assignment** 
*$\alpha :\Delta \to \prod_{\delta \in \Delta }S\left(\delta \right)$ provides a value α(δ)∈S(δ) to each face δ∈Δ. A* 
**partial assignment***, β, provides a value for a subset $\Delta^{\prime }\subset \Delta$ of faces, $\beta \Delta^{\prime }\prod_{\delta \in \Delta^{\prime }}S\left(\delta \right)$. An assignment s is called a* 
**global section** 
*if for each inclusion δ⇝λ of faces, S(δ⇝λ)(s(δ))=s(λ).*


As an example, a possible global section for our tracking sheaf is:$$\begin{array}{c} s(\mathrm{Text})=‘\mathrm{Intersection}\;\mathrm{of}\;\mathrm{Victoria}\;\mathrm{Rd}\;\mathrm{and}\;\mathrm{Meadow}\;\mathrm{Rd}’\\ s\left(\mathrm{Vehicle}\mathrm{GPS}\right)=\left(\begin{array}{c} 35.6^{\circ }\;\mathrm{lat}\\ -82.6^{\circ }\;\mathrm{long}\\ 2019\;\mathrm{ft} \end{array}\right),\\ s\left(\mathrm{Bear}\mathrm{Collar}\right)=\left(\begin{array}{c} 358391\;E\\ 3936750\;N\\ 581\;m \end{array}\right),\\ s\left(\mathrm{Radio}\mathrm{VHF}\mathrm{Device}\right)=\left(\begin{array}{c} 358943\;E\\ 3936899\;N\\ 615.4\;m\\ 572\;m\\ 195^{\circ } \end{array}\right),\\ s\left(\left\{\mathrm{Radio}\mathrm{VHF}\mathrm{Device},\mathrm{Bear}\mathrm{Collar}\right\}\right)=\left(\begin{array}{c} 358391\;E\\ 3936750\;N \end{array}\right)\\ \mathrm{and}\;s(\{\mathrm{Text},\mathrm{Vehicle}\mathrm{GPS}\})=s(\{\mathrm{Text},\mathrm{Radio}\mathrm{VHF}\mathrm{Device}\})\\ =s(\{\mathrm{Vehicle}\mathrm{GPS},\mathrm{Radio}\mathrm{VHF}\mathrm{Device}\})\\ =s\left(\left\{\mathrm{Text},\mathrm{Vehicle}\mathrm{GPS},\mathrm{Radio}\mathrm{VHF}\mathrm{Device}\right\}\right)=\left(\begin{array}{c} 358943\;E\\ 3936899\;N\\ 615.4\;m \end{array}\right). \end{array}$$

### 4.2. Consistency Structures, Pseudosections, and Approximate Sections

At a given time, a global section of our tracking model corresponds to the respective sensor readings simultaneously agreeing on the location of both the researcher and the bear. This would be ideal, however, in practice it is very unlikely that all the sensor readings will agree to the precision shown above. For certain applications, such as our tracking model, the equality constraint of a global section may be too strict. Consistency structures [71] and the consistency radius [2] address this concern. Consistency structures loosen the constraint that sensor data on the vertices match, on passing through restrictions, by instead requiring that they merely *agree*. Agreement is measured by a boolean function on each individual face. The pairing of a sheaf with a collection of these boolean functions is called a **consistency structure**.

**Definition** **4.***A* 
***consistency structure** 
*
*is a triple (Δ,S,C) where *Δ* is an abstract simplicial complex, S is a sheaf over* Δ, *and C is the assignment to each non-vertex d-face λ∈Δ,d>0, of a function*
Cλ:S(λ)dimλ+1→{0,1}
*where Zk denotes the set of multisets of length k over Z.*


The multiset in the domain of $C_{\lambda }$ represents the multiple sheaf values to be compared for all the vertices impinging on a non-vertex face λ, while the codomain {0,1} indicates whether they match “well enough” or not. In particular, we have the **standard consistency structure** for a sheaf S, which assigns an *equality* test to each non-vertex face $\lambda =\{v_{1},v_{2},\cdots ,v_{k}\}$:$$C_{\lambda }\left(\left[z_{1},z_{2},\cdots ,z_{k}\right]\right)=\left\{\begin{array}{cc} 1, & \mathrm{if}\;z_{1}=z_{2}=\cdots =z_{k}\\ 0, & \mathrm{otherwise}\; \end{array}\right.$$
where [·] denotes a multiset, and $z_{i}=S\left(\left\{v_{i}\right\}⇝\lambda \right)\left(\alpha \left(\left\{v_{i}\right\}\right)\right)$.

A consistency structure broadens the equality requirement natively available in a sheaf to classes of values which are considered equivalent. However, beyond that, in our tracking model, each stalk is a metric space. Thus we can use the natural metric to test that points are merely “close enough”, rather than agreeing completely or being equivalent. Using ϵ to indicate the quantitative amount of error present or tolerated in an assignment, we define the **ϵ-approximate consistency structure** for a sheaf S as follows. For each non-vertex size *k* face $\lambda =\{v_{1},v_{2},\cdots ,v_{k}\}\in \Delta ,k>1$ define
$$C_{\lambda }\left(\left[z_{1},z_{2},\cdots ,z_{k}\right]\right)=\left\{\begin{array}{cc} 1 & \mathrm{if}\;\hat{\sigma }\left(\left[z_{1},z_{2},\cdots ,z_{k}\right]\right)\leq \epsilon ,\\ 0 & \mathrm{otherwise},\; \end{array}\right.$$
where
$$\hat{\sigma }\left(Y\right)=\sqrt{\frac{1}{\left|Y\right|}\sum_{y\in Y}||y-\mu_{Y}{\left|\right|}^{2}}=\sqrt{\frac{1}{\left|Y\right|}\mathrm{Tr}\left(\Sigma_{Y}\right)}$$

$\mu_{Y}$ is the mean of the set *Y*, and $\Sigma_{Y}$ is the covariance matrix of the multidimensional data *Y*. This measure of consistency gives a general idea of the spread of the data.

The corresponding notion of a global section for sheaves is called a pseudosection for a consistency structure.

**Definition** **5.** 
*An assignment $s\in \prod_{\delta \in \Delta }S\left(\delta \right)$ is called a (Δ,S,C)-*
* 
**pseudosection** 
*
*if for each non-vertex face $\lambda =\{v_{1},\cdots ,v_{k}\}$*


*$C_{\lambda }\left(\left[S\left(\left\{v_{i}\right\}⇝\lambda \right)s\left(\left\{v_{i}\right\}\right)\;:\;i=1,\cdots ,k\right]\right)=1$, and*

*$C_{\lambda }\left(\left[S\left(\left\{v_{i}\right\}⇝\lambda \right)s\left(\left\{v_{i}\right\}\right)\;:\;i=1,\cdots ,j-1,j+1,\cdots k\right]\cup \left[s\left(\lambda \right)\right]\right)=1$ for all j=1,2,⋯,k.*



An assignment *s* which is a pseudosection guarantees that for any non-vertex face λ, (1) the restrictions of its vertices to the face are close, and (2) the value assigned to the face is consistent with the restricted vertices.

When C is the standard consistency structure, then it comes about that
$$s\left(\lambda \right)=S\left(v_{1}⇝\gamma \right)s\left(\left\{v_{1}\right\}\right)=S\left(v_{2}⇝\gamma \right)s\left(\left\{v_{2}\right\}\right)=\cdots =S\left(v_{k}⇝\gamma \right)s\left(\left\{v_{k}\right\}\right)$$
for each non-vertex face $\lambda =\{v_{1},\cdots ,v_{k}\}$. Thus pseudosections of a standard consistency structure are global sections of the corresponding sheaf.

On the other hand, when C is the ϵ-approximate consistency structure, then we have that
$\hat{\sigma }\left(\left[S\left(\left\{v_{i}\right\}⇝\lambda \right)s\left(\left\{v_{i}\right\}\right)\;:\;i=1,\cdots ,k\right]\right)\leq \epsilon$, and$\hat{\sigma }\left(\left[S\left(\left\{v_{i}\right\}⇝\lambda \right)s\left(\left\{v_{i}\right\}\right)\;:\;i=1,\cdots ,j-1,j+1,\cdots k\right]\cup \left[s\left(\lambda \right)\right]\right)\leq \epsilon$ for all j=1,2,⋯,k.
for each non-vertex face $\lambda =\{v_{1},v_{2},\cdots ,v_{k}\}$.

Pseudosections of ϵ-approximate consistency structures are precisely the generalization needed for our tracking model. The ϵ-approximate consistency structure over our tracking sheaf consists of the following assignment of functions:CX,CY,CZ:R32→{0,1},CB:R22→{0,1}CH:R33→{0,1}
where
$$C_{X}\left(\left[z_{1},z_{2}\right]\right)=C_{Y}\left(\left[z_{1},z_{2}\right]\right)=C_{Z}\left(\left[z_{1},z_{2}\right]\right)=C_{B}\left(\left[z_{1},z_{2}\right]\right)=\left\{\begin{array}{c} 1\;\mathrm{if}\;\hat{\sigma }\left(\left[z_{1},z_{2}\right]\right)\leq \epsilon \\ 0\;\mathrm{otherwise}\; \end{array}\right.$$
and
$$C_{H}\left(\left[z_{1},z_{2},z_{3}\right]\right)=\left\{\begin{array}{c} 1\;\mathrm{if}\;\hat{\sigma }\left(\left[z_{1},z_{2},z_{3}\right]\right)\leq \epsilon \\ 0\;\mathrm{otherwise}.\; \end{array}.\right.$$

A pseudosection for this consistency structure is an assignment
⟨s(R),s(v),s(T),s(G),s(X),s(Y),s(Z),s(B),s(H)⟩
such that for each non-vertex face λ∈{X,Y,Z,B,H}

$\hat{\sigma }\left(\left[S\left(\left\{v\right\}⇝\lambda \right)s\left(\left\{v\right\}\right)\;:\;v\in \lambda \right]\right)\leq \epsilon$, and$\hat{\sigma }\left(\left[S\left(\left\{v\right\}⇝\lambda \right)s\left(\left\{v\right\}\right)\;:\;v\in \lambda \right]\cup \left[s\left(\lambda \right)\right]\backslash \left[s\left(w\right)\right]\right)\leq \epsilon$ for all w∈λ.

Now s(R), s(V), s(T), and s(G) are measurements given by the data feeds, so we are left to assign s(X), s(Y), s(Z), s(B), and s(H) to minimize ϵ. Physically, a pseudosection of the ϵ-approximate consistency structure for our tracking model assigns positions for both the human and bear so that the “spread” of all measurements attributed to any face is bounded by ϵ. The question then becomes how does one minimize ϵ efficiently? The theorem below states that the minimum ϵ for which a pseudosection exists is determined only by the restricted images of the vertices.

**Theorem** **1.**
*Let S be a sheaf over an abstract simplicial complex *Δ* such that each stalk is a metric space and let $s\in \prod_{\delta \in \Delta }S\left(\delta \right)$ be an assignment. The minimum ϵ for which s is a pseudosection of the ϵ-approximate consistency structure (Δ,S,C) is*
$$\epsilon^{*}=\max_{\lambda \in \Delta \backslash \{\{v\}v\in V\}}\hat{\sigma }\left(\left[S\left(\left\{w\right\}⇝\lambda \right)s\left(\left\{w\right\}\right)w\in \lambda \right]\right).$$


We call this minimum error value $\epsilon^{*}$ the **consistency radius**.

The consistency radius is a quantity that is relatively easy to interpret. If it is small, it means that there is minimal disagreement among observations. If it is large, then it indicates that at least some sensors disagree.

The proof of this theorem is an application of the following Lemma.

**Lemma** **1.**
*Let $Z=\{z_{1},\cdots ,z_{k}\}$ be a set of real numbers with mean $\mu_{Z}$. For z∈Z, define $Y_{z}=Z\backslash z\cup \mu_{Z}$. Then $\forall z\in Z,\hat{\sigma }\left(Y_{z}\right)\leq \hat{\sigma }\left(Z\right)$.*


**Proof.** Fix x∈Z. Please note that $|Y_{z}|=k$, and that $\hat{\sigma }\left(Y_{z}\right)\leq \hat{\sigma }\left(Z\right)$ if and only if $\hat{\sigma }{\left(Y_{z}\right)}^{2}\leq \hat{\sigma }{\left(Z\right)}^{2}$. Next,
$$\hat{\sigma }{\left(Y_{z}\right)}^{2}=\frac{1}{k}\sum_{i=1}^{k}{\left(y_{i}-\mu_{Y}\right)}^{2}\leq \frac{1}{k}\sum_{i=1}^{k}{\left(y_{i}-\mu_{Z}\right)}^{2}$$
since the expression is minimized around the mean. Then
$$\frac{1}{k}\sum_{i=1}^{N}{\left(y_{i}-\mu_{Z}\right)}^{2}\leq \hat{\sigma }{\left(Z\right)}^{2}$$
as $0\leq {\left(z-\mu_{Z}\right)}^{2}$, completing the proof.

### 4.3. Maximal Consistent Subcomplexes

The association of data with a signal network, as in the case of our tracking model, lends itself naturally to the question of consistency. Data is received from various signal sources, some of which should agree but may not. Thus, one would like to identify maximal consistent portions of the network. In [72], Praggastis gave an algorithm for finding a unique set of maximally consistent subcomplexes.

Let (Δ,S,C) be a consistency structure. For each τ∈Δ, we define star(τ)={ρ∈Δ:τ⊆ρ} to be the set of faces containing τ. We can think of the star of a face as its sphere of influence. Similarly, for $\Delta^{\prime }\subset \Delta$, we define $\mathrm{star}\left(\Delta^{\prime }\right)=\underset{\tau \in \Delta^{\prime }}{⋃}\mathrm{star}\left(\tau \right)$. Next, for a subset of vertices W⊆U, we define $\Delta_{W}=\left\{\tau \in \Delta \;:\;\tau \subseteq W\right\}$ to be the subcomplex of Δ induced by *W*. Please note that $\Delta_{W}$ inherits the consistency structure $\left(\Delta_{W},S\left(\Delta_{W}\right),C{|}_{\Delta_{W}}\right)$ from (Δ,S,C). Now let *s* be a sheaf partial assignment on just the vertices *U*. We say that *s* is consistent on $\Delta_{W}$ if $C_{\gamma }\left(\left[S\left(v⇝\gamma \right)s\left(v\right)\;:\;v\in \gamma \right]\right)=1$ for each non-vertex face γ in $\Delta_{W}$. The theorem below states that we can associate a unique set of maximally consistent subcomplexes to any vertex assignment.

**Theorem** **2.** 
*Let (Δ,S,C) be a consistency structure and let α be a sheaf partial assignment on U. There exists a unique collection of subsets $\left\{W_{i}\right\}$ of U which induce subcomplexes $\left\{\Delta_{W_{i}}\right\}$ of *Δ* with the following properties:*


*The assignment α is consistent on each $\Delta_{W_{i}}$, and any subcomplex on which α is consistent has some $\Delta_{W_{i}}$ as a supercomplex.*
*$⋃\mathrm{star}(\Delta_{W_{i}})$ is a cover of* Δ.



The proof of this result is constructive. The basic operation is described in the following lemma.

**Lemma** **2.** 
*Let (Δ,S,C) be a consistency structure and let α sheaf partial assignment on U. If $C_{\gamma }\left(\left[S\left(v⇝\gamma \right)\alpha \left(v\right)\;:\;v\in \gamma \right]\right)=0$ for some non-vertex face γ∈Δ, then there exist subsets $\left\{W_{i}\right\}$ of U which induce subcomplexes $\left\{\Delta_{W_{i}}\right\}$ of *Δ* such that:*


*$\gamma \notin \Delta_{W_{i}}$ for all i.*

*Every subcomplex on which α is consistent has some $\Delta_{W_{i}}$ as a supercomplex.*

*$⋃\mathrm{star}(\Delta_{W_{i}})$ is a cover of Δ.*



**Proof.** For each vertex $v_{i}\in \gamma$, let $W_{i}=U\backslash v_{i}$ with induced subcomplex $\Delta_{W_{i}}$. Clearly γ does not belong to any of the $\Delta_{W_{i}}$, and if $\Delta^{\prime }$ is a subcomplex of Δ on which α is consistent, then the vertices of $\Delta^{\prime }$ is a subset of some $W_{i}$. Finally, since every vertex in *U* belongs to at least one $W_{i}$, we have that $\Delta =⋃\mathrm{star}(\Delta_{W_{i}})$.

**Proof of** **Theorem 2.**If α is consistent on Δ, we are done, so suppose that $C_{\gamma }\left(\left[S\left(v⇝\gamma \right)\alpha \left(v\right)\;:\;v\in \gamma \right]\right)=0$ for some non-vertex face γ∈Δ. By Lemma 2, there exists a set of subcomplexes $\left\{\Delta_{W_{i}}\right\}$ such that (1) $\gamma \notin \Delta_{W_{i}}$ for all *i*, (2) every subcomplex on which α is consistent has some $\Delta_{W_{i}}$ as a supercomplex, and (3) $⋃\mathrm{star}(\Delta_{W_{i}})$ is a cover of Δ. Next, we repeat the process for each complex $\Delta_{W_{i}}$ individually. If α is inconsistent on $\Delta_{W_{i}}$, we apply Lemma 2 and replace $W_{i}$ with $\left\{W_{i_{k}}\right\}$ and $\Delta_{W_{i}}$ with $\left\{\Delta_{W_{i_{k}}}\right\}$.Since Δ is finite, the process must terminate with a list of consistent subcomplexes. It is possible to have nested sequences of subcomplexes in the final list. Since we are only interested in maximal consistent subcomplexes, we drop any complex which has a supercomplex in the list. Let $\left\{W_{i}\right\}$ and $\left\{\Delta_{W_{i}}\right\}$ be the final list of vertex sets and subcomplexes, respectively, obtained in this way. By construction, α is consistent on each $\Delta_{W_{i}}$. Further, by Lemma 2, any subcomplex on which α is consistent belongs to some $\Delta_{W_{i}}$ and the set $\left\{\mathrm{star}\left(\Delta_{W_{i}}\right)\right\}$ is a cover of Δ. Finally, suppose that $\left\{X_{j}\right\}$ and $\left\{\Delta_{X_{j}}\right\}$ are another pair of vertex sets and subcomplexes which satisfy the properties of the theorem. Fix some $\Delta_{X_{k}}$. Since α is consistent on $\Delta_{X_{k}}$, there exists some $\Delta_{W_{\ell }}$ such that $\Delta_{X_{k}}\subseteq \Delta_{W_{\ell }}$. However, by reversing the argument, there is also some $\Delta_{X_{m}}$ such that $\Delta_{W_{\ell }}\subseteq \Delta_{X_{m}}$. Finally, since $\Delta_{X_{k}}$ is maximal in $\left\{\Delta_{X_{j}}\right\}$, we must have that k=m and $\Delta_{X_{k}}=\Delta_{W_{\ell }}$. ☐

In the context of our tracking model, given a collection of sensor readings and an ϵ, Theorem 2 provides a maximal vertex cover so that each associated subcomplex is ϵ-approximate consistent.

### 4.4. Measures on Consistent Subcomplexes

In this section, we define a measure on the vertex cover associated with a set of maximal consistent subcomplexes by identifying the set of covers as a graded poset. We use the rank function of the poset as a measure. For a comprehensive background on posets, the interested reader is referred to [73].

Let P=〈P,≤〉 be a poset. For a subset X⊆P, let ↓X={s≤x:x∈X} denote the ideal of *X* in *P*. The set of all ideals, I(P), ordered by inclusion, forms a poset denoted J(P). Additionally, there is a one-to-one correspondence between J(P) and the set of all antichains A(P). Specifically, the functions
$$\begin{array}{cc} \max & :J(P)\to A(P),\;\max(I)=\{x\in I\;:\;x\neg <y\;\mathrm{for}\;\mathrm{all}\;y\in I\},\;\mathrm{and}\;\\ ↓ & :A(P)\to J(P),\;↓(A)=↓A \end{array}$$
are inverses of each other. Next, recall that a poset is called graded if it can be stratified.

**Definition** **6.***A poset P=〈P,≤〉 is* 
***graded** 
*
*if there exists a rank function r:P→N∪{0} such that r(s)=0 if s is a minimal element of P, and r(q)=r(p)+1 if p≺q in P. If r(s)=i, we say that s has rank i. The maximum rank, $\max_{p\in P}\left\{r\left(p\right)\right\}$, is called the*
* 
**rank** 
*
*of P.*


As a consequence of the Fundamental Theorem for Finite Distributive Lattices, it is well known that J(P) is a graded poset.

**Proposition** **1.**
*[73] If P=〈P,≤〉 is an n-element poset, then J(P) is graded of rank n. Moreover, the rank r(I) of I∈J(P) is the cardinality of I.*


Now consider $2^{n}$, the Boolean lattice on *n* elements. Explicitly, $2^{n}$ consists of all subsets of {1,2,⋯,n} where for all a,b⊆{1,2,⋯,n} we define a≤b precisely when a⊆b, and the lattice operations are defined as a∧b=a∩b, and a∨b=a∪b. The vertex cover $\left\{W_{i}\right\}$ associated with a maximal set of consistent subcomplexes $\left\{\Delta_{W_{i}}\right\}$, provided by Theorem 2, can be viewed as an antichain of $2^{n}$ where n=|U|. Moreover, by design, each of these antichains is a set cover of {1,2,⋯,n}. We will call such an antichain **full**. Specifically, $A=\{a_{1},a_{2},\cdots ,a_{k}\}$ is said to be full if ${⋃}_{i=1}^{k}a_{i}=\left\{1,2,\cdots ,n\right\}$. The set of all full ideals $\bar{I(2^{n})}$ forms an induced subposet $\bar{J(2^{n})}$ of the graded poset $J(2^{n})$. Next, we show that $\bar{J(2^{n})}$ is also graded.

**Proposition** **2.**
*$\bar{J(2^{n})}$ is a graded poset of rank $2^{n}-\left(n+1\right)$.*


**Proof.** By Proposition 1, $J(2^{n})$ is a graded poset of rank $2^{n}$. Moreover, the rank function is given by r(I)=|I|. Now $\bar{J(2^{n})}$ is an induced subposet of $J(2^{n})$ with the single minimum element 0:={∅,{1},{2},⋯,{n}}. Define the function
$$\bar{r}:\bar{J(2^{n})}\to N\;\cup \;\left\{0\right\},\;\bar{r}\left(I\right)=r\left(I\right)-r\left(0\right)=\left|I\right|-\left(n+1\right).$$Please note that if $I_{1}≺I_{2}$ in $\bar{J(2^{n})}$, then $I_{1}≺I_{2}$ in $J(2^{n})$. The fact that $\bar{r}$ is a rank function then follows from the fact that *r* is a rank function. ☐

Since $\bar{J(2^{n})}$ is a graded poset, we can use the rank function as a measure on the vertex cover associated a set of maximal consistent subcomplexes.

**Definition** **7.**
*Let (Δ,S,C) be a consistency structure with |V|=n and let $A=\{W_{i}\}$ be a vertex cover obtained from Theorem 2. Then we define*
$$\bar{r}\left(A\right)=\left|↓A\right|-\left(n+1\right).$$


Please note that $0\leq \bar{r}\left(A\right)\leq 2^{n}-\left(n+1\right)$ and a larger value indicates more consistent subcomplexes. Indeed, if $↓A_{1}\subseteq ↓A_{2}$, then $\bar{r}\left(A_{1}\right)\leq \bar{r}\left(A_{2}\right)$.

### 4.5. Consistency Filtrations

Now, consider a simplicial complex Δ, a sheaf S on Δ, and partial assignment α to the vertices of Δ. Given any value ϵ≥0 we can consider the ϵ-approximate consistency structure $\left(\Delta ,S,C_{\epsilon }\right)$ and use Theorem 2 to obtain the set of maximal consistent subcomplexes for the assignment α. Varying ϵ we obtain a filtration, which we call the **consistency filtration**, of vertex covers corresponding to landmark ϵ values $\epsilon_{0}=0<\epsilon_{1}<\cdots <\epsilon_{\ell -1}<\epsilon_{\ell }=\epsilon^{*}$ where we recall that $\epsilon^{*}$ is the consistency radius, i.e., the smallest value of ϵ for which α is a $\left(\Delta ,S,C_{\epsilon }\right)$-pseudosection. The corresponding refinement of vertex covers is $C_{0}\leq C_{1}\leq \cdots \leq C_{\ell -1}\leq C_{\ell }=U$ where each is a set of subcomplexes whose union is Δ. We can also compute the sequence of cover measures $p_{0}<p_{1}<\cdots <p_{\ell -1}<p_{\ell }=1$ for each $C_{i}$. (A somewhat different perspective on the consistency filtration is discussed in [51], in which the consistency filtration is shown to be both functorial and robust to perturbations, and is so in both the sheaf and the assignment.)

The consistency filtration is a useful tool for examining consistency among a collection of sensors. If the distance between two particular consecutive landmark values $\epsilon_{i}$ and $\epsilon_{i+1}$ is considerably larger than the rest, this indicates that there is considerable difficulty in resolving a disagreement between at least two groups of sensors. Which sensors are at fault for this disagreement can be easily determined by comparing the covers $C_{i}$ and $C_{i+1}$.

## 5. Results

As detailed in Section 3, data were gathered for twelve locating sessions, six with dummy collars hidden around the city and six with live bears. The six live sessions were distributed over three female bears with session frequencies 3, 2, and 1. Each locating session was two hours long and included 16 measurements.

We first present overall measurement results across both bear and dummy collars. We then drill down to a particular bear, N024, in the context of its overall consistency radius. We conclude with a detailed examination of a single measured assignment to this bear, particularly at minute 5.41, in order to understand the details of the consistency filtration.

### 5.1. Overall Measurements

Overall consistency radii $\epsilon^{*}$ are shown below: Figure 4 shows for the full system; Figure 5 for the human; and Figure 6 for the bears.

The most obvious trend in Figure 4 is a decrease in consistency radius over time. This implies that the overall error in the measurements of the bear-human system decrease over time. Because the consistency radius for the measurements of the humans alone (Figure 5 does not show this trend while the measurements of bear alone (Figure 6) does, this implies that there is a disagreement between the various measurements of the bear, but not the humans. This indicates that there is an overall improvement in agreement between measurements as the experiment progresses. In fact, as subsequent sections will show, the VHF measurements degrade with increasing range. As the experiment progressed, the humans approached the location of the bear, and so the error in the VHF measurements decreased. As the humans approach the bear, this means that the VHF measurements are are more consistent with the other measurements. This tends to reduce the consistency radius since small consistency radius means that the data and model are in agreement.

On the other hand, the reader is cautioned that consistency radius does not and cannot tell you *what* the measurements are, nor does it tell you which measurements are more trustworthy. However, the consistency radius does not presuppose any distributions, nor does it require any parameter estimation in order to indicate agreement. As the next few sections indicate, the consistency filtration does a good job of blaming the sensor at fault for the time variation seen in Figure 4 and Figure 5, again with minimal assumptions. Strictly speaking, trying to use the sensors to estimate some additional quantity is an aggregation step that brings with it additional assumptions. Using the consistency radius alone avoids making these assumptions.

The reader may ask why considering the time variation of consistency radius is more useful than some aggregate statistic, such as its maximum value. However, taking the max (or any other extremal statistic) is poor statistical practice, and worthy of being avoided on those grounds! Timeseries are almost always better at expressing system behaviors, and in our case it happens to reveal a clear trend.

### 5.2. Example: Bear N024

We now consider in detail the readings for Bear N024, recorded on 6 January 2016. First we show (some of) the raw data readings. Figure 7 shows a sample of the first twelve data points for the car GPS *V*, showing latitude, longitude, and altitude. Figure 8 shows the readings from the bear GPS *B*, including UTM and elevation. Finally, Figure 9 shows a combination of the text reading *T* and the radiocolar *R*, including the five variables UTM, elevation, and bearing and range to the bear. Ranges in increments of meters according to Table 3.

Since the sensors read out on different time scales, interleaving of the data records is required, as illustrated in Figure 10 for the first 18 integrated readings, encompassing 909 s. Figure 11 shows a similar record, now compacted and sampled at the minute interval, showing repeated entries for sensors across all time stamps. A map is shown in Figure 12, with the red dots showing the position of the bear’s GPS collar, and the blue dots and lines showing the measured positions of the human, including V,T, and *R*. Finally, the time sequence of the consistency radius is shown in Figure 13, where data points are adorned with the addresses visited over time, on a general basis, along with a linear trendline.

### 5.3. Example: Minute 5.41

We now explicate the concept of a consistency filtration over a single set of measurements for N024 bear, specifically the point from minute 5.41, when the sensor readings shown on the left side of Figure 14 were recorded, as shown in the sheaf model. After this raw data is processed through the sheaf model functions shown in Figure 3, the resulting processed sensor readings are then shown on the right side of Figure 14, along with the resulting spread variance (meters) measures shown on non-vertex faces in blue.

The consistency radius for this measurement is $\epsilon^{*}=464.5$; this is the maximum variance recorded among the sensors, in particular between the receiver *R* and the GPS on the bear *G*. Our consistency parameter will range over $\epsilon \in \left[0,\epsilon^{*}\right]=\left[0,464.5\right]$ over a series of steps, or landmark values, 1≤i≤ℓ of the filtration. At each step the following occur:There is a landmark non-zero consistency value $0<\epsilon_{i}\leq \epsilon^{*}$ which does not exceed the consistency radius;There is a prior set of consistent faces $\Gamma_{i-1}$;A new consistent face γ∈Δ is added so that $\Gamma_{i}=\Gamma_{i-1}\left\{\gamma \right\}$;There is a corresponding vertex cover $\Lambda_{i}$, which is a coarsening of the prior $\Lambda_{i-1}$;And which has a cover measure $\bar{r}\left(\Lambda \right)$.

These steps for our example are summarized in Table 4, and explained in detail below.

$\epsilon_{0}=0$:If we insist that no error be tolerated, that is that all data be consistent, then any nontrivial set in the vertex cover, produced by Theorem 2, cannot contain nontrivial faces of Δ. As such, the set of consistent faces are just the singletons $\Gamma_{0}=\left\{\left\{V\right\},\left\{R\right\},\left\{T\right\},\left\{G\right\}\right\}$, and the vertex cover is $\Lambda_{0}=\{\left\{T,G\right\},\left\{V,G\right\}$, {R}} with $\bar{r}\left(\Lambda_{0}\right)=2/11$.$\epsilon_{1}=9.48$:If we relax our error threshold to the next landmark value, while still well below our consistency radius, the readings on *V* and *R* are considered consistent within this tolerance, so that the face Y={V,R} is added, yielding the new set of consistent faces
$$\Gamma_{1}=\Gamma_{0}\left\{Y\right\}=\left\{\left\{V\right\},\left\{R\right\},\left\{T\right\},\left\{G\right\},\left\{V,R\right\}\right\}.$$The new vertex cover is $\Lambda_{1}=\left\{\left\{V,R\right\},\left\{V,G\right\},\left\{G,T\right\}\right\}$, with cover measure $\bar{r}\left(\Lambda_{1}\right)=3/11$.$\epsilon_{2}=15.9$:Continuing on, next *T* and *R* come into consistency, adding the face Z={R,T}, yielding
$$\Gamma_{2}=\left\{\left\{V\right\},\left\{R\right\},\left\{T\right\},\left\{G\right\},\left\{V,R\right\},\left\{T,R\right\}\right\},$$
$$\Lambda_{2}=\left\{\left\{V,R\right\},\left\{V,G\right\},\left\{G,T\right\},\left\{R,T\right\}\right\},\;\bar{r}\left(\Lambda_{2}\right)=4/11.$$$\epsilon_{3}=18.42$:The next landmark introduces the three-way interaction H={V,T,R} (for notational simplicity just note that $\Gamma_{3}=\Gamma_{2}\left\{H\right\}$). However, the vertex cover is unchanged, yielding $\Lambda_{3}=\Lambda_{2}$ and $\bar{r}\left(\Lambda_{3}\right)=\bar{r}\left(\Lambda_{2}\right)=4/11$.$\epsilon_{4}=20.35$:Next *T* and *V* are reconciled, adding X={V,T} to $\Gamma_{4}$. Now $\Lambda_{4}=\left\{\left\{V,T,R\right\},\left\{V,T,G\right\}\right\}$, with $\bar{r}\left(\Lambda_{4}\right)=7/11$.$\epsilon_{5}=\epsilon^{*}=464.5$:Finally we arrive at our consistency radius with the bear collar *G* being reconciled to *R* adding the face {B,G} to $\Gamma_{5}$. Our vertex cover is naturally now the coarsest, i.e., just the set of vertices $\Lambda_{5}=\left\{\left\{V,T,R,G\right\}\right\}$ as a whole, with $\bar{r}\left(\Lambda_{5}\right)=1$.

These results are summarize in Table 4, while Figure 15 shows a plot of the cover coarseness as a function of consistency, and adorned by the *combination* of sensors engaged at each landmark. It is evident that the most discrepency, by far, among sensors comes from the Radio VHF Device and Bear Collar when reporting on the bear’s location.

Figure 15 shows the filtration for a single time stamp, t=5.41, with each of the five sensor combinations possible in the simplicial complex appearing at a certain point in the filtration, with, in particular, the RG interaction for the bear sensors highlighted. Figure 16 shows these across all time steps for bear N024. We can now see that RG isn’t always the least consistent sensor readings, with, at times, VR and VT also leaping to the top position over several time intervals.

## 6. Comparison Statistical Model

Sheaf modeling generally, and especially as introduced in this paper, is a novel approach to multi-sensor fusion and target tracking. As such, it behooves us to consider alternate, or standard, approaches which analysts may bring to bear on the question of both predicting locations and quantifying uncertainty around them. For these purposes, we have also performed in parallel a statistical model of the bear tracking data. Specifically, a dynamic linear model (DLM) is introduced, whose parameters are then estimated using a Kalman filter. A comparison of these results with the sheaf model is instructive as to the role that sheaf modeling can play in the analytical landscape as *generic* integration models.

Although we perform a comparison between these two approaches, we suggest to the reader that these two approaches are complementary. Consistency radius and filtration do not and cannot tell what the measurements are, nor does it report which measurements are more trustworthy. So while we show that comparison with a Kalman filter’s covariance estimates is apt, comparing consistency radius with the Kalman filter’s position estimates is not appropriate. The Kalman filter provides an additional aggregation stage to the process, but one that also brings additional assumptions along with it. If those assumptions are not valid, the process may produce faulty results.

A DLM is a natural statistical approach for representing the bear/human tracking problem. A DLM separately models the movements of the bear, movements of the human, and the multiple observations on each. These model components are combined based on a conditional independence structure that is standard for hidden Markov models [74]. Once stochastic models of the movements and of the observations are obtained, then multiple standard computational approaches are available to use the model and the data to:Estimate the locations, with corresponding uncertainties, over time of the bear collar and the human (see Figures 18 and 19 below).Estimate accuracy parameters of the involved sensors based on each of the single runs of the experiment (see Table 5 below).Combine information across the multiple experimental runs to estimate the accuracy of the sensors (see Equation (5) below).

The Kalman Filter (KF) [75,76] is a standard computational approach for estimation with a DLM from which the above three items can be computed. Monte Carlo approaches for estimating parameters in DLMs are also available [77]. In addition, this type of model is also called State Space modeling [75], and Data Augmentation - depending on the technical community. Similar tracking approaches are given in numerous references [3,78,79].

### 6.1. High-Level State and Observation Equations - With Examples

The two sets of equations below are a schematic for the DLM that supports bear and human tracking. First - the state equations that represent the bear and human locations are:$$\begin{array}{cc} {\mathrm{bear}}_{t+\Delta t} & ={\mathrm{bear}}_{t}+\omega_{b,\Delta t,t+\Delta t}\\ {\mathrm{human}}_{t+\Delta t} & ={\mathrm{human}}_{t}+\omega_{h,\Delta t,t+\Delta t} \end{array}$$

The state equations represent both the locations and the movements of the bear and the human over time. In this particular set of state equations the locations of each of the bear and human are modeled as random walks that are sampled at non-equispaced times. The fact that Δt here is not constant is an important sampling feature: the measurements are not all taken at equal time intervals. The parameters to be set, and/or estimated are embedded in the two ω terms. These terms are stochastically independent Gaussian random variables. The covariance matrices for the ω’s are parameters to be set or estimated. Intuitively, a larger variance corresponds with potentially more rapid movement. Finally, these state equations are motivated by those used for general smoothing in [80].

The schematic observation equations are:$$\begin{array}{cc} \mathrm{truck}\;{\mathrm{gps}}_{t} & ={\mathrm{human}}_{t}+\epsilon_{\mathrm{truck}\;\mathrm{gps},t}\\ \mathrm{vhf}\;{\mathrm{gps}}_{t} & ={\mathrm{human}}_{t}+\epsilon_{\mathrm{vhf}\;\mathrm{gps},t}\\ {\mathrm{streetsign}}_{t} & ={\mathrm{human}}_{t}+\epsilon_{\mathrm{streetsign},t}\\ \mathrm{bear}\;{\mathrm{gps}}_{t} & ={\mathrm{bear}}_{t}+\epsilon_{\mathrm{bear}\;\mathrm{gps},t}\\ {\mathrm{vhf}}_{t} & ={\mathrm{human}}_{t}-{\mathrm{bear}}_{t}+\epsilon_{\mathrm{vhf},t} \end{array}$$

These equations connect directly with the state equations, and represent a full set of observations at time *t*. If, as is typical for this data (see Figure 10), there is not a complete set of information at a particular time, then the observation equations are reduced to reflect only those measurements that have been made. The KF calculations for estimating bear/human locations along with sensor accuracies can proceed with that irregularly sampled data. The computations for collar N024 are in two dimensions (see Figures 18 and 19 below), but for conciseness the equations shown for the KF are in one dimension.

A general form for a DLM that shows the update from time t−Δt to time *t* is: (2)$$\begin{array}{ccc} \theta_{t} & = & G_{\Delta t}\theta_{t-\Delta t}+\omega_{t} \end{array}$$(3)$$\begin{array}{ccc} y_{t} & = & H_{t}\theta_{t}+\epsilon_{t}\;\;t=1,2,3,\ldots \end{array}$$
where $G_{\Delta t}$ is an identify matrix. The base state model is a random walk – and so the ‘‘best guess’’ is centered on the previous location. The covariance of $\omega_{t}$ is structured as a diagonal matrix with diagonals proportional to the elapsed time Δt. The parameters that are needed to be chosen to calculate a DLM are the variances of the ’impetus’ $\omega_{t}$ (which depends on Δt)and the variances of the observation errors $\epsilon_{t}$. The model assumes that the means and correlations of these are zero. The one-dimensional bear equations map to this abstract representation as:$$\begin{array}{cc} \theta_{t} & =\left(\begin{array}{c} {\mathrm{bear}}_{t}\\ {\mathrm{human}}_{t} \end{array}\right)\\ y_{t} & =\left(\begin{array}{c} \mathrm{truck}\;{\mathrm{gps}}_{t}\\ \mathrm{vhf}\;{\mathrm{gps}}_{t}\\ {\mathrm{streetsign}}_{t}\\ \mathrm{bear}\;{\mathrm{gps}}_{t}\\ {\mathrm{vhf}}_{t} \end{array}\right) \end{array}$$
with corresponding expansions for $\omega_{t}$ and $\epsilon_{t}$.

The data are structured so that it rarely if ever occurs that all 5 measurements are taken at the same time. For a particular time at which an observation is available the observation equations are subsetted down to the observation or observations that are available. This can be seen in the ’time interweave’ plot for one of the experiment runs. This is represented in the general form of the DLM below (Equations (Equation 2) and (Equation 3)) by varying the matrix $H_{t}$ with corresponding variations in $\epsilon_{t}$. For complete observations at time *t* the matrix $H_{t}$ and $\epsilon_{t}$ would respectively be:(4)$$\begin{array}{c} H_{t}=\left(\begin{array}{cc} 0 & 1\\ 0 & 1\\ 0 & 1\\ 1 & 0\\ -1 & 1 \end{array}\right),\;\epsilon_{t}=\left(\begin{array}{c} \epsilon_{\mathrm{truck}\;\mathrm{gps},t}\\ \epsilon_{\mathrm{vhf}\;\mathrm{gps},t}\\ \epsilon_{\mathrm{streetsign},t}\\ \epsilon_{\mathrm{bear}\;\mathrm{gps},t}\\ \epsilon_{\mathrm{vhf},t} \end{array}\right) \end{array}$$

The variance parameters are:$$\begin{array}{c} \sigma =\left(\begin{array}{c} \sigma_{\mathrm{truck}\;\mathrm{gps}}\\ \sigma_{\mathrm{vhf}\;\mathrm{gps}}\\ \sigma_{\mathrm{streetsign}}\\ \sigma_{\mathrm{bear}\;\mathrm{gps}}\\ \sigma_{\mathrm{vhf}} \end{array}\right) \end{array}$$

Correspondingly, if only the VHF data are available then $H_{t}$ and $\epsilon_{t}$ would respectively be:$$\begin{array}{c} \left(\begin{array}{cc} -1 & 1 \end{array}\right)\;\left(\begin{array}{c} \epsilon_{\mathrm{vhf},t} \end{array}\right), \end{array}$$
and if only the bear collar GPS and truck GPS data are available then $H_{t}$ and $\epsilon_{t}$ would respectively be:$$\begin{array}{c} \left(\begin{array}{cc} 1 & 0\\ -1 & 1 \end{array}\right)\;\left(\begin{array}{c} \epsilon_{\mathrm{bear}\;\mathrm{gps},t}\\ \epsilon_{\mathrm{vhf},t} \end{array}\right). \end{array}$$

### 6.2. Estimation of Parameters

The parameters are estimated via the Kalman filter. That calculation is representable as a Bayesian update as follows. At time t−1 the knowledge of the state parameter, informed by all the data up to and including that time, is represented as:$${\hat{\theta }}_{t-1}|y_{t-1}∼N\left({\hat{\theta }}_{t-1},\Sigma_{t-1}\right)$$

This equation is read as ’the distribution of ${\hat{\theta }}_{t-1}$ given the data $y_{t-1}$ is Gaussian with a mean of ${\hat{\theta }}_{t-1}$ and a covariance matrix $\Sigma_{t-1}$’. The Kalman filter is typically written in two steps: a forecast followed by an update. With $e_{t}=y_{t}-H_{t}G_{\Delta t}{\hat{\theta }}_{t-1}$, $V_{t}$ the covariance of $\epsilon_{t}$, and $W_{t}$ the covariance of $\omega_{t}$, then the forecast step is:$${\hat{\theta }}_{t}|y_{t-1}∼N\left(G_{\Delta t}{\hat{\theta }}_{t-1},R_{t}=G_{\Delta t}\Sigma_{t-1}G_{\Delta t}^{\prime }+W_{t}\right)$$
and the update step is:$${\hat{\theta }}_{t}|y_{t}∼N\left(G_{\Delta t}{\hat{\theta }}_{t-1}+R_{t}H_{t}^{\prime }\left(V_{t}+H_{t}R_{t}H_{t}^{\prime }\right)e_{t},R_{t}-R_{t}H_{t}^{\prime }{\left(V_{t}+H_{t}R_{t}H_{t}^{\prime }\right)}^{-1}H_{t}R_{t}\right)$$

The above steps are executed sequentially by passing through the data items in time order.

The parameters in the DLM may be chosen as values which maximize the likelihood function. The set of variance parameters includes those for the five observation forms and the state equations—we denote this collection by σ. The values σ can be estimated from the data using as the values that maximize the *likelihood function*. The likelihood function is
$$L\left(y_{1},\ldots ,y_{n}∥\sigma \right)=\prod_{i=1}^{n}p\left(y_{i}∥\sigma ,y_{i-1},\ldots ,y_{1}\right)$$

The terms in the product are available as a byproduct of a KF calculation. Table 5 shows the estimated parameters for collar N024. These were estimated from the collar data, using the KF computation and the maximum likelihood approach. As expected, the estimated parameter for the VHF antenna measurement is larger than the other observation parameters. Also, the human (in the vehicle) is seen to move more quickly than the bear. The table shows the parameters as standard deviations so that the units are interpretable.

The KF structure supports pooling the models and measurements across the 12 experiments. This has the potential to provide more informed estimates of the sensor errors (as represented by the variances) compared with the single experiment based estimates shown in Table 5. The information from these series can be pooled to arrive at estimates of σ under the assumption that the variances that correspond with the five observations are the same across experiments. In that case the pooled likelihood function is:(5)$$\prod_{j=1}^{12}L_{j}\left(y_{j}∥\sigma \right)$$

In addition, the estimates of σ can be estimated as those which maximize the pooled likelihood. Another case is to take advantage of the series corresponding with the cases where the bear collar was stationary by setting that variance component to zero in the state update. The remaining variances can be estimated for those series with that assumption—essentially reducing the dimensionality of the optimization problem.

### 6.3. Example Outputs

Figure 17 is a map showing output for the DLM for the human position plotted against the measured position, with the KF estimated human positions shown in red, and the direct measurements of human position shown in blue. Figure 18 then shows example output for the location estimates of the bear, and Figure 19 for the human. These plots also show estimated 95% confidence intervals for the locations. Human position refers to the estimated human position from the sensors at specific times. Both the sheaf and the KF output location estimates for the bear and the human. The x-axis is time(in seconds) from the first observation. The estimates for bear locations are shown as green circles for both the ’east’ and ’north’ coordinates. The location units are UTM on the y-axis. The observed locations for the bear collar are from the GPS measurements. Two standard deviation intervals that were estimated are shown as gray lines. The implementation of the KF that we used was set to output a location estimate plus uncertainty for each time that an observation in the overall system was made. The jumps in the estimated values are seen to primarily shift when a bear collar GPS observation is made. Jumps seen elsewhere correspond with large shifts in the related data - which in this case includes the VHF measurement and its link with the human’s position.

The estimates for human location are also shown in green, gps data for the human (truck gps) in blue, and confidence intervals in gray. Again - there are more data than shown in the plot, so the jumps in the estimated locations that do not directly correspond with data points on the plots are driven by other data in the data set.

### 6.4. Comparisons

The *processed* sheaf model and the dynamic linear model of our example capture the same phenomenon, and even do so in relatively similar ways. Because of the canonicity of the sheaf modeling approach, any other sensor integration model must recapitulate aspects of the sheaf model. After loading processed data into the sheaf, the state space that governs the dynamic linear model is precisely the space of global sections of the sheaf model. If we do not register the text reports or the angular measurements from the VHF receiver into a common coordinate system, then while the sheaf model is still valid, the data are no longer related linearly. Without this preprocessing, the dynamic linear model is not valid, and we conclude that the sheaf carries more information. Therefore, all of the following comparisons are made considering only observations registered into a common coordinate system.

The observation equations of the dynamic linear model (represented by the matrix $H_{t}$) can therefore be “read off” the restriction maps of the sheaf. The two models differ more in choices made by the modelers: (1) the sheaf model treats uncertainty parametrically, while the dynamic linear model does so stochastically, (2) the sheaf model processes time slices independently, without a dynamical model. We address these two differences in some detail after first addressing their similar treatment of the state space and observational errors.

While the space of assignments for the sheaf S defined in our example is quite large, the space of global sections is comparatively small. It is parameterized by the four real numbers, specifically the east and north positions of the human and the bear. This is precisely the space of state variables $\theta_{t}$ at each timestep. On the other hand, the $y_{t}$ matrix of observations consists of values within an assignment ot the vertices. The observation matrix $H_{t}$ is an aggregation of all of the corresponding restriction maps, deriving the values of a global section at each vertex. The observational errors $\epsilon_{t}$ are then the differences between the observations we actually made (an assignment), and those predicted from the state variables (a global section). Thus the consistency radius is merely the largest of the compoents of $\epsilon_{t}$.

Figure 20 is a map showing sheaf bear location estimates in purple and the DLM and bear location estimates in red. Figure 21 shows the human sheaf location estimates in purple and the measured locations for humans in blue. The most significant difference between the proposed sheaf-based model and dynamic linear model of sensor-to-sensor relationships is their treatment of uncertainty. The dynamic linear model assumes that the data are subject to a stochastic model, which determine independent Gaussian random variables for the positions of the bear and human. The sheaf model posits a deterministic model for these independent variables, but permits (and quantifies) violations of this model. The two models are in structural agreement on how to relate groups of non-independent variables. Both models assert exact agreement of the three observations of the human position, and that the VHF observations ought to be the vector difference between bear and human position. Specifically, the observation matrix $H_{t}$ (Equation (4)) and the sensor matrix determining the base space for the sheaf (Table 1) are identical up to sign (the difference in signs is due to differences in the choice of basis only). In particular, the vector difference for relating the VHF observations to bear and human positions is encoded in the restriction map S(R⇝B) shown in the right frame of Figure 3 and also in the last row of the $H_{t}$ matrix.

While the two models treat relationships between variables identically, we have chosen to treat the individual sensors differently with respect to time. The sheaf model is not intended to estimate sensor *self* consistency directly. Sensor self consistency can only be estimated in the sheaf by way of comparison against other sensors. In contrast, the dynamic linear model works through time (Equations (Equation 2) and (Equation 3)) to estimate self consistency from the sensor’s own time series. Indeed, the dynamic linear model separates observational errors $\epsilon_{t}$ from those arising from impetus $\omega_{t}$, which can be thought of as deviations from the baseline, and can estimate relative sensor precisions. It should be noted that this distinction could be made using a sheaf by explicitly topologizing time, though this was not the focus of our study.

Considering the specific case of Collar N024, the difference between the VHF-estimated and GPS-estimated bear positions determines the consistency radius. This is shown by the presence of RG  being the largest value in Figure 16 for most time values. Because of this, baseline comparisons of the observational error the VHF sensor computed by the dynamic linear model in Table 5 broadly agrees with the typical consistency radius shown in Figure 13.

Focusing on the specific time discussed in Section 5.3, namely *t* = 5.41 min = 325 s as shown in Figure 15, it is clear from the sheaf analysis that the majority of the error is due to the difference between the VHF observation and the bear collar GPS. These inferences are not easily drawn from the dynamic linear model output in Figure 18 and Figure 19, but the difference between the quality of these sensors is shown in Table 5.

Delving more finely into the timeseries, the consistency radius for the bear-human system (Figure 13) shows a steady decrease as the human approaches the bear, leading to improved VHF readings. This is also visible in Figure 18, in which the error variance steadily decreases for the bear location estimates.

The three sensors on the human show more subtle variations, as is clear from the orange curve on the left side of Figure 5 and Figure 19. In both, there is a marked increase in human error near *t* = 75 min = 4500 s. Consulting the consistency filtration in Figure 15, this is due primarily to the consistency radius being determined by the edge VR  in the base space of the sheaf—a difference between the GPS receiver on the vehicle and the GPS receiver attached to the VHF receiver. Shorter occurrences of the same phenomenon occur near times *t* = 15 min = 900 s and *t* = 32 min = 1920 s.

The dynamic linear model posits that changes in the state are governed by an impetus vector $\omega_{t}$, while our sheaf model is essentially not dynamic. While it may seem that these two models are at variance with one another, they are easily rectified with one another. Considering only the global sections of S, which is the state space of the dynamic linear model, we build a sheaf over a line graph, namely in which *I* represents the identity matrix. Global sections of this sheaf consist of timeseries of the internal state $\theta_{t}$ with no impetus, while an assignment to the vertices corresponds to a timeseries of internal state *with* impetus. Thus again, the consistency radius measures the maximum impetus present.

## 7. Conclusions

Sheaves are a mathematical data structure that can assimilate heterogeneous data types, making them ideal for modeling a variety of sensor feeds. Sheaves over abstract simplicial complexes provide a global picture of the sensor network highlighting multi-way interactions between the sensors. As a consequence of the foundational nature of sheaves, we know that *any* multi-sensor fusion method can, in principle, be encoded as a sheaf. Additionally, the theory provides algorithms for assessing measurement consistencies. In addition, in this article, we demonstrated how one might use sheaves for tracking wildlife with a collection of sensors collecting types of information. We saw that sheaves can be used to produce a holistic temporal picture of the observations, and can describe the levels of agreement between different groups of sensors.

Since our approach is quite general, it could be applied to many other use cases. Perhaps the most obvious application is that of locating emergency beacons from downed aircraft [2] or other hidden radio transmitters [81]. More broadly, the idea of consistency radius can be valuable in combining disparate biochemical networks [82], analyzing the convergence of graphical models and numerical differential equation solvers [42,45], and estimating network flows [58,59,83].

The generality and expressivity of a sheaf-based approach, therefore, is to be valued, albeit in the context of consideration of its costs, potential limitations, and comparison with other methods. For example, when we compared our sheaf model with standard tracking algorithms, the sheaf model was able to provide error measurements without any upfront parameter estimation. For instance, sheaves and dynamic linear modeling encode dependence between observations starting from the same information—a relation between sensors and state variables. While broadly in quantitative agreement on our example data, these two methods format this information differently. These differences result in certain inferences being easier to make in one or the other framework. For instance, because dynamic linear models require parameter estimation, sensor self-consistency is easier to compute as a consequence. Conversely, since the sheaf model does not require parameter estimation, this simplifies the process of making fine-grained inferences about which groups of sensors provide consistent observations.

So while sheaf models obviate the need for up-front uncertainty quantification, on the other hand they carry the burden of up-front model construction and complexity. In particular, the specific sensor architecture, including detailed knowledge about pairwise sensor interactions, need to be encoded in the abstract simplicial complex. Additionally, the ability to model all of the resulting complex interactions itself carries an additional computational burden: while the initial model setup requires specification of *pairwise* sensor interactions, the resulting sheaf model calculates *all* multi-way interactions through the abstract simplicial complex.

Finally, it should be noted that the sheaf-based approach can be extended to handle targets that move at higher speeds. For instance, [50] demonstrates that the same recipe for constructing a sheaf as presented here (with different sensors: passive RF sensors and optical cameras) can be used to track moving vehicles under tree cover. Their resulting sheaf-based tracker smoothly handles sensor hand-off and maintains track custody even when the target is occluded.

## Figures and Tables

**Figure 1 sensors-20-03418-f001:**
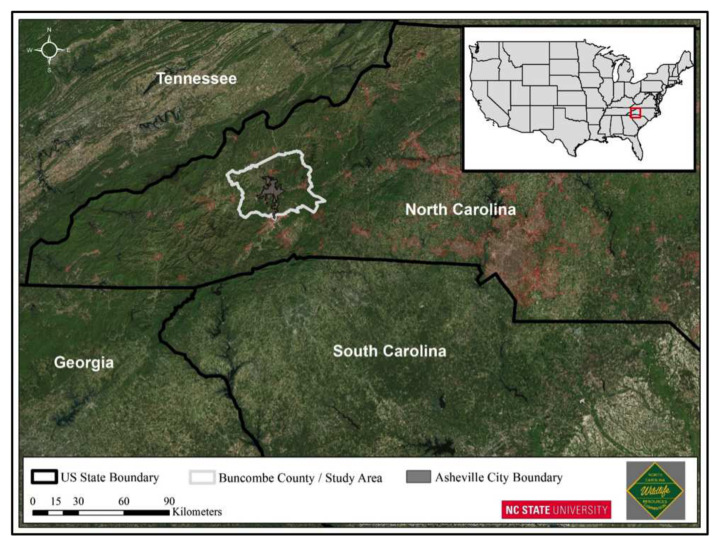
NCSU/NCWRC urban/suburban black bear (Ursus americanus) study area, Asheville, North Carolina, USA

**Figure 2 sensors-20-03418-f002:**
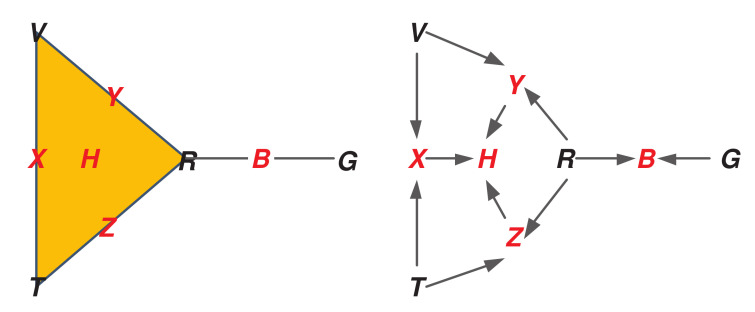
(**Left**) Simplicial complex of tracking sensors. (**Right**) Attachment diagram.

**Figure 3 sensors-20-03418-f003:**
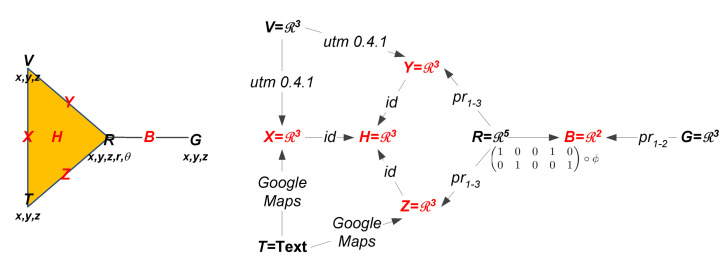
(**Left**) ASC adorned with stalks on the sensor vertices. (**Right**) Sheaf model.

**Figure 4 sensors-20-03418-f004:**
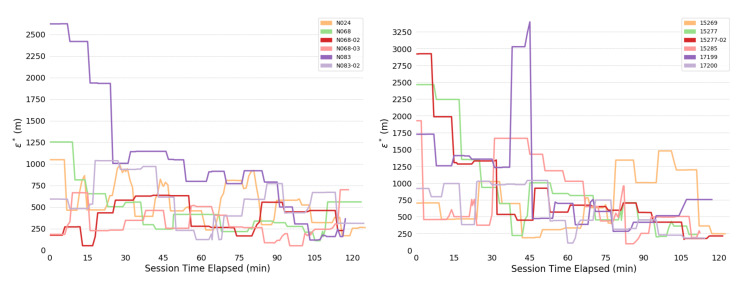
Consistency radii $\epsilon^{*}$ over time, full system. (**Left**) Bears. (**Right**) Dummy collars.

**Figure 5 sensors-20-03418-f005:**
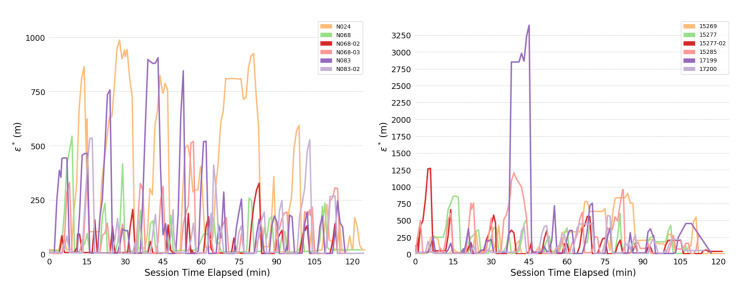
Consistency radii $\epsilon^{*}$ over time, full system. (**Left**) Bears. (**Right**) Dummy collars.

**Figure 6 sensors-20-03418-f006:**
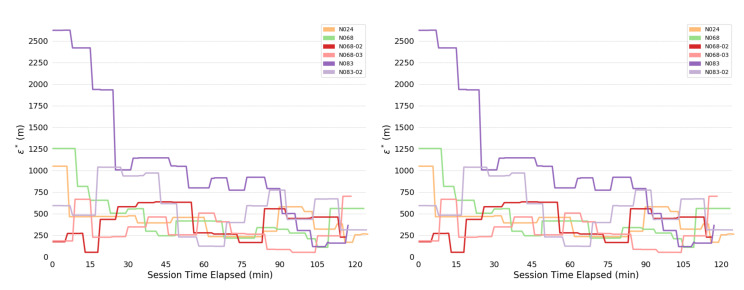
Consistency radii $\epsilon^{*}$ over time, human. (**Left**) Bears. (**Right**) Dummy collars.

**Figure 7 sensors-20-03418-f007:**
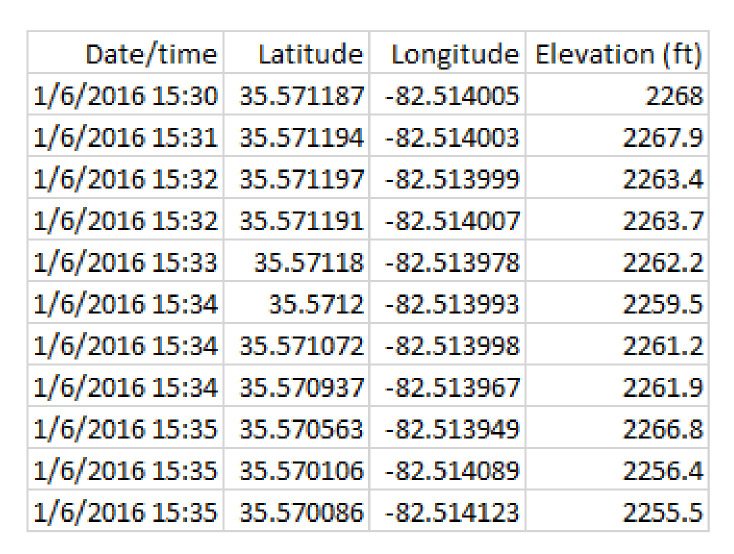
Vehical GPS readings *V* for Bear N024.

**Figure 8 sensors-20-03418-f008:**
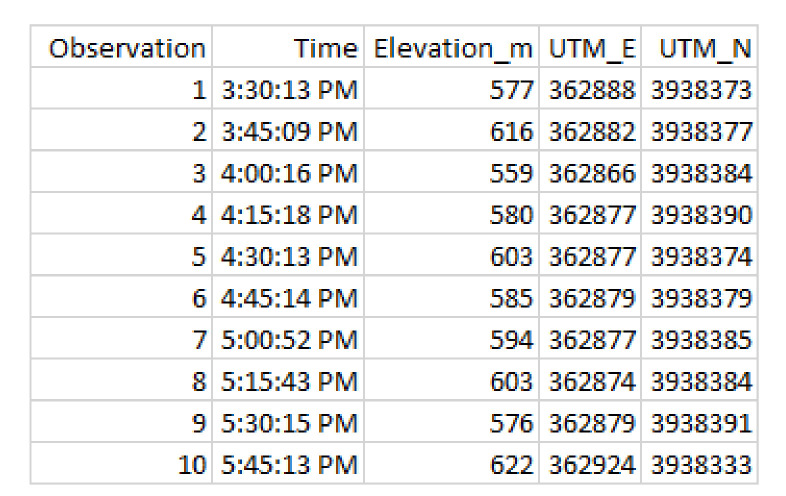
Bear position *B* for Bear N024.

**Figure 9 sensors-20-03418-f009:**
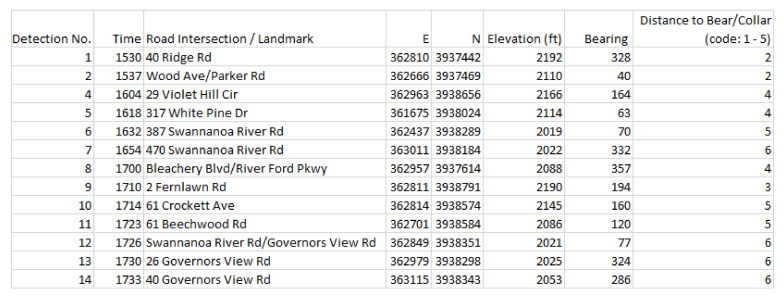
Readings for text location *T* and radio receiver *R* for Bear N024.

**Figure 10 sensors-20-03418-f010:**
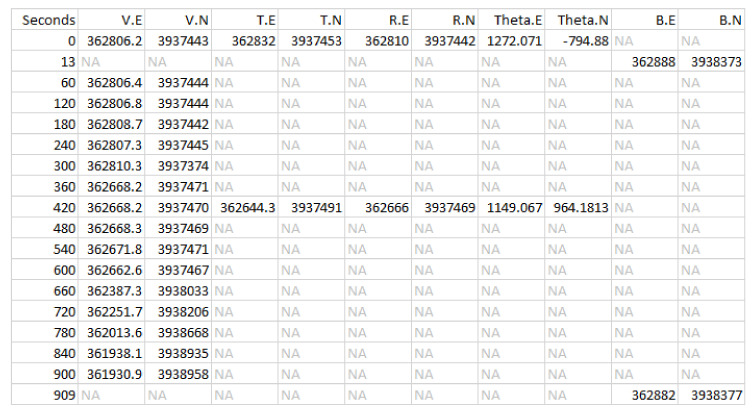
Time-integrated sensors streams. Variables V,T,R, and *B* as in the model, with Theta being the offset to the bear on the VHF receiver (see Equation (Equation 1)), x.N being northing, and x.E being easting in UTM.

**Figure 11 sensors-20-03418-f011:**
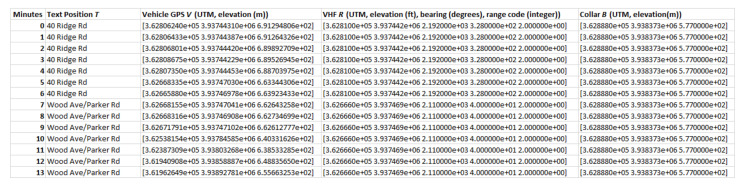
Combined data record for collar N024 interpolated to the minute.

**Figure 12 sensors-20-03418-f012:**
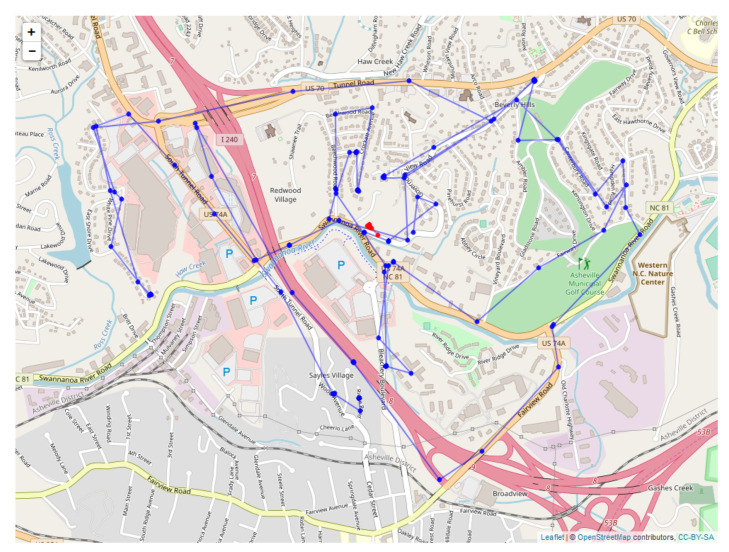
Map of sensor readings for bear N024.

**Figure 13 sensors-20-03418-f013:**
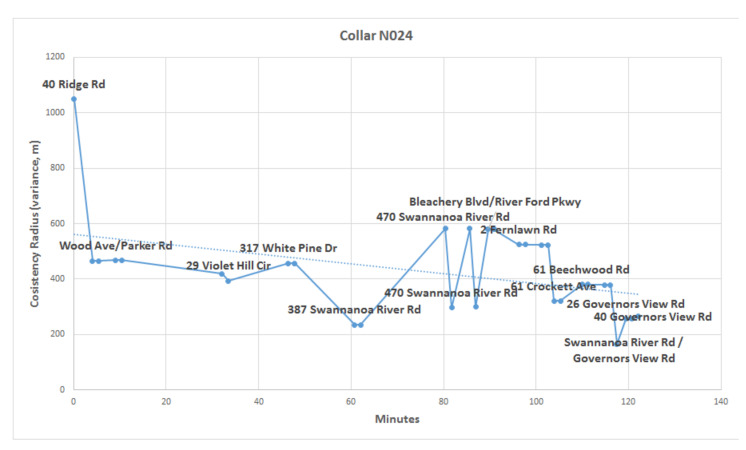
Consistency radius $$\epsilon^{*}$$ over time for bear N024.

**Figure 14 sensors-20-03418-f014:**
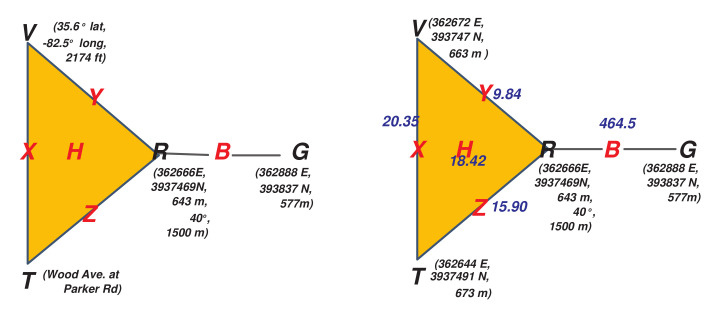
(**Left**) A raw assignment on collar N024. (**Right**) Processed data and spread measures
(variance in m, in blue).

**Figure 15 sensors-20-03418-f015:**
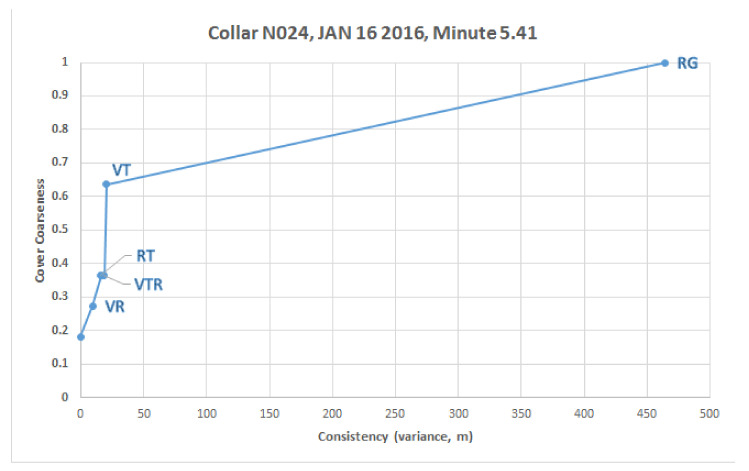
(**Left**) A raw assignment on collar N024. (Consistency vs. cover coarseness for example
assignment.

**Figure 16 sensors-20-03418-f016:**
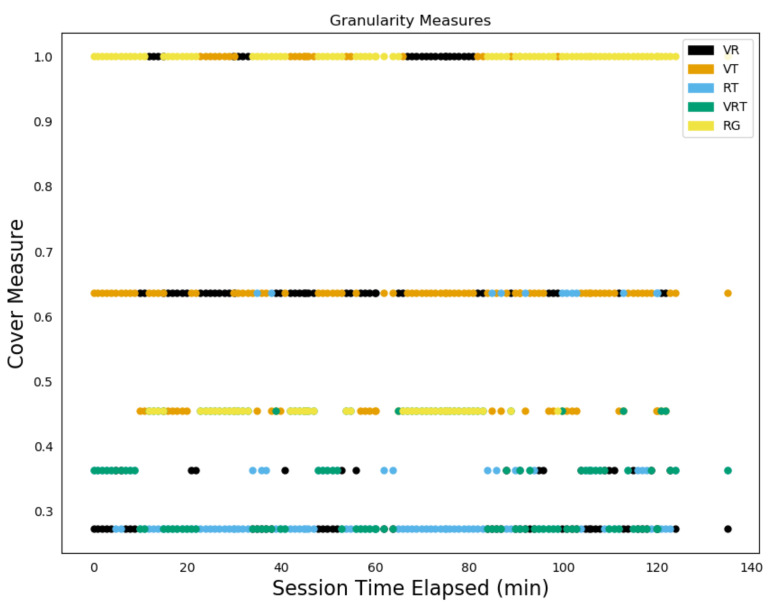
Consistency filtration values for entire N024 trajectory.

**Figure 17 sensors-20-03418-f017:**
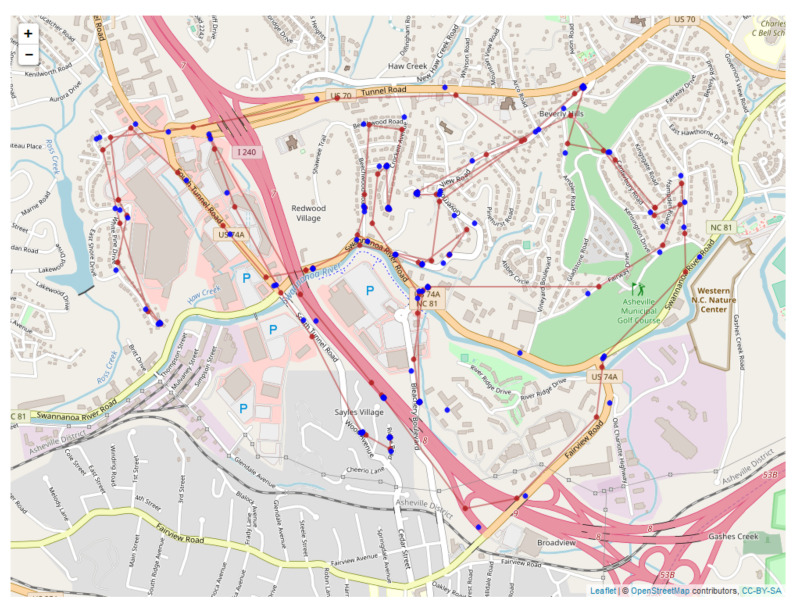
Output for the DLM for the human position (red) plotted against the measured position
(blue).

**Figure 18 sensors-20-03418-f018:**
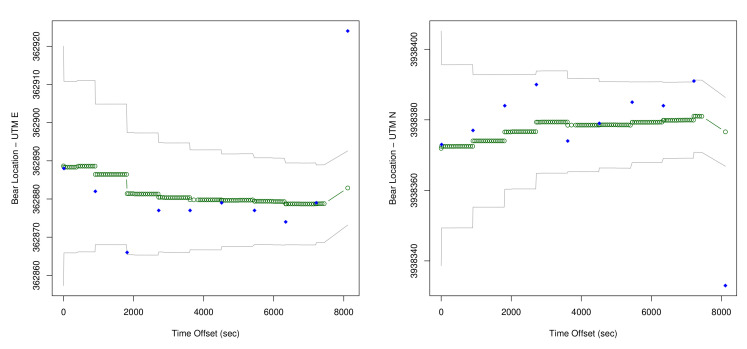
KF Estimates of locations of the bear for N024. (Estimates of location are shown in green,
the GPS data is shown in blue, and the confidence intervals are shown in gray.)

**Figure 19 sensors-20-03418-f019:**
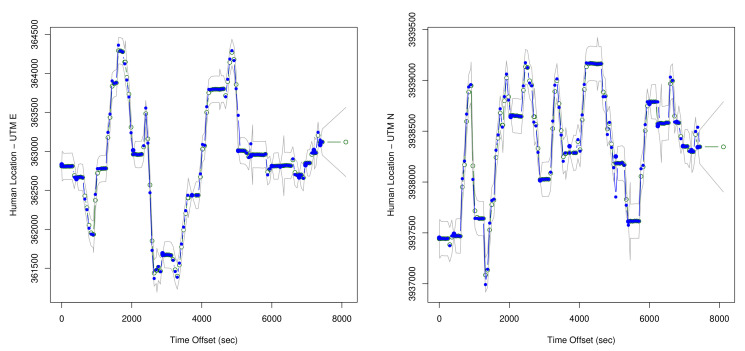
KF Estimates of locations of thehuman for N024. (Estimates of location are shown in green,
the GPS data is shown in blue, and the confidence intervals are shown in gray.)

**Figure 20 sensors-20-03418-f020:**
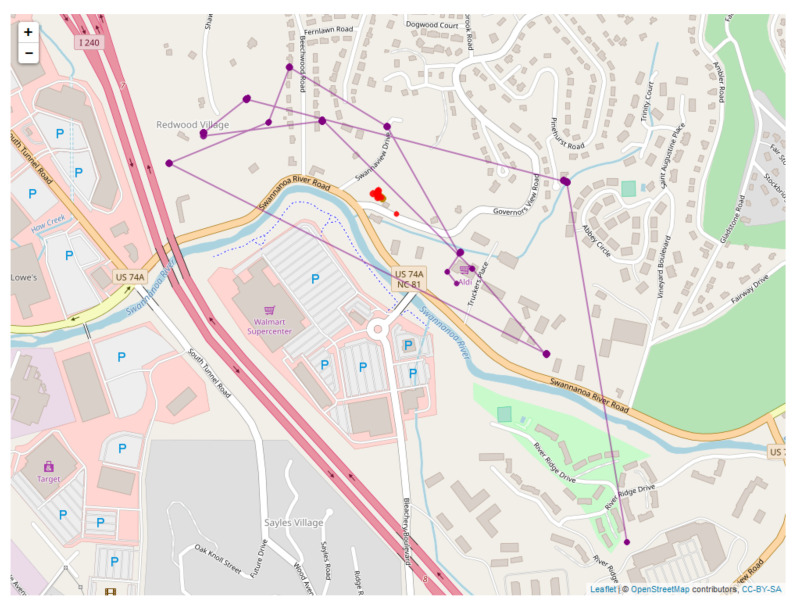
Comparison of the DLM (red) and sheaf model (purple) location estimates for the bear.

**Figure 21 sensors-20-03418-f021:**
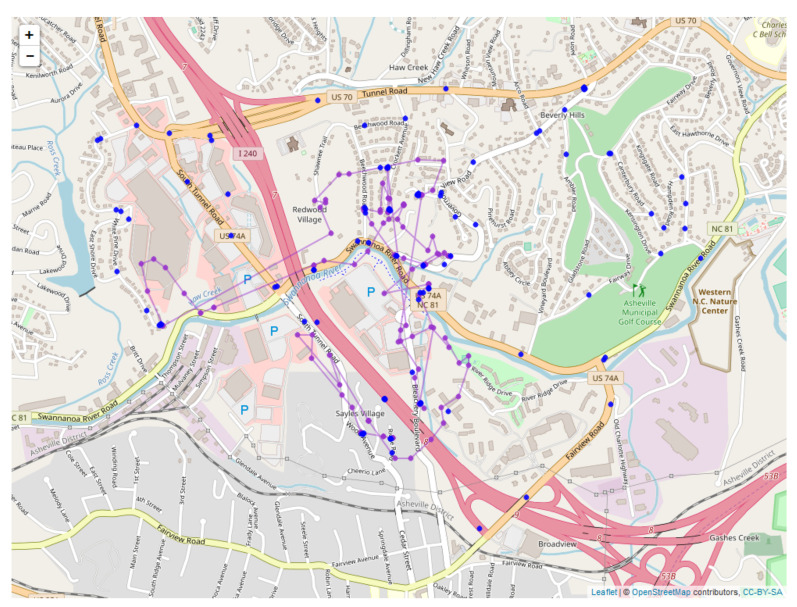
Comparison of the sheaf (purple) and measured data (blue) for the human location.

**Table 1 sensors-20-03418-t001:** Sensor matrix for the bear tracking.

	H= Human Position	B= Bear Position
V= Vehicle GPS ⟨lat,long,ft⟩	*√*	
T= Text	*√*	
R= Receiver ⟨UTM N,UTM E,m,deg,m⟩	*√*	*√*
*G* = GPS on Bear ⟨UTM N, UTM E,m⟩		*√*

**Table 2 sensors-20-03418-t002:** Data feeds for tracking sheaf model.

Vertex	Data Format	Description	Stalk
G= *Bear Collar*	(E, N, m)	Position and elevation of bear from collar	$R^{3}$
*V*=*Vehicle GPS*	(lat, long, ft)	Position and elevation of human from vehicle	$R^{3}$
T= *Text*	string	Text description of human’s location	set of strings
R= *Radio VHF Device*	(E, N, m, m, deg)	Position and elevation of human and position of bear relative to human	$R^{5}$

**Table 3 sensors-20-03418-t003:** Ranges in meters for each distance code.

Distance Code	Distance (m)
2	1500
3	1000
4	750
5	500
6	375

**Table 4 sensors-20-03418-t004:** Consistency filtration of our example assignment.

ϵ	New Consistent Face	Vertex Cover	Measure
0.00		{{T,G},{V,G},{R}}	2/11
9.48	Y={V,R}	{{V,R},{V,G},{G,T}}	3/11
15.90	Z={R,T}	{{V,R},{V,G},{G,T},{R,T}}	4/11
18.42	H={V,T,R}	{{V,R},{V,G},{G,T},{R,T}}	4/11
20.35	X={V,T}	{{V,T,R},{V,T,G}}	7/11
464.50	B={R,G}	{{V,T,R,G}}	1

**Table 5 sensors-20-03418-t005:** Estimated KF parameters for collar N024. The first two rows are the standard deviation estimates for the ω parameters. The remaining rows contain standard deviation estimates for the ϵ parameters.

Parameter Standard Deviation	Value (m)
Bear state update	0.008
Human state update	26.524
Bear GPS Obs.	16.091
VHF GPS Obs.	0.032
Vehicle GPS Obs.	91.434
Street sign Obs.	27.597
VHF Obs.	663.998

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
