# Peer review of "A Sheaf Theoretical Approach to Uncertainty Quantification of Heterogeneous Geolocation Information"

_sensors, 2020, doi:10.3390/s20123418_

Round 1

Reviewer 1 Report

The authors presented a very well written paper that can potentially be used to address challenges associated with combining observations from different sensor types to track multiple targets.

The reviewer has two minor suggestions:

  1. What are the limitations of the paper or what are the limitations of the proposed method? The authors should put a few sentences in the conclusions to highlight its limitations and address how potential future research can address these limitations. 
  2. The authors can potentially strengthen their literature review by adding a few recently published papers in the transportation domain where heterogeneous sensors are often used. In addition, most of the papers that the authors cited are before 2015 and more recent publications can be added to reflect the development in the field. Based on a quick Google Scholar search, there are a few suggestions:

Zhu, S., Guo, Y., Chen, J., Li, D., & Cheng, L. (2017). Integrating Optimal Heterogeneous Sensor Deployment and Operation Strategies for Dynamic Origin-Destination Demand Estimation. Sensors17(8), 1767.

Hurley, J. D., Johnson, C., Dunham, J., & Simmons, J. (2018, April). Multilevel probabilistic target identification methodology utilizing multiple heterogeneous sensors providing various levels of target characteristics. In Signal Processing, Sensor/Information Fusion, and Target Recognition XXVII (Vol. 10646, p. 106461M). International Society for Optics and Photonics.

Yue, Y., Yang, C., Wang, Y., Senarathne, P. C. N., Zhang, J., Wen, M., & Wang, D. (2019). A multilevel fusion system for multirobot 3-d mapping using heterogeneous sensors. IEEE Systems Journal.

The authors should consider added additional recent publications to strengthen their literature review section. 

Author Response

  1. “What are the limitations of the paper or what are the limitations of the proposed method? The authors should put a few sentences in the conclusions to highlight its limitations and address how potential future research can address these limitations. 

We have included additional discussion in the conclusion, Section 7, lines 754-787. While we have generally edited that section, the key new lines include:

“we know that {\em any} multi-sensor fusion method can, in principle, be encoded as a sheaf.”

“The generality and expressivity of a sheaf-based approach, therefore, is to be valued, albeit in the context of consideration of its costs, potential limitations, and comparison with other methods.”

“So while sheaf models obviate the need for up-front uncertainty quantification, on the other hand they carry the burden of up-front model construction and complexity. In particular, the specific sensor architecture, including detailed knowledge about pairwise sensor interactions, need to be encoded in the abstract simplicial complex. Additionally, the ability to model all of the resulting complex interactions itself carries an additional computational burden: while the initial model setup requires specification of {\em pairwise} sensor interactions, the resulting sheaf model calculates {\em all} multi-way interactions through the abstract simplicial complex.”

  1. The authors can potentially strengthen their literature review by adding a few recently published papers in the transportation domain where heterogeneous sensors are often used. In addition, most of the papers that the authors cited are before 2015 and more recent publications can be added to reflect the development in the field. Based on a quick Google Scholar search, there are a few suggestions:

Thank you, we very much appreciate the recent citations, and have examined them, and introduced some comments on them in Section 2.2, lines 109-114 and 132-142.

Reviewer 2 Report

Dear Authors,

The paper is overall well written and easy to follow. The provided analysis of the model is concrete and correct.

The proposed approach is very interesting and the benefits in enabling data fusion and tracking might have huge impacts to several domains.

I would suggest you to clarify more and more the following points:

1) Do you envision any limitations for tracking high speed targets (e.g., autonomous cars with high frequency sampling rate) and to adopt the system for tracking flying systems in free air (e.g., UAV)?

2) Can you confirm that the homology of your simplicial complex will be always  Betti_0 = 1, Betti_i = 0 for i >= 1 ? Or can you clarify this aspect? It might suggest further investigations.

3) I suggest you to add in the conclusion a detailed list of other potential use cases that would benefit of your approach. 

Author Response

1) Do you envision any limitations for tracking high speed targets (e.g., autonomous cars with high frequency sampling rate) and to adopt the system for tracking flying systems in free air (e.g., UAV)?

Thank you, the following text has been inserted in Section 7 lines 783-787.

The sheaf-based approach can be extended to handle targets that move at higher speeds.  For instance, \cite{robinson2018dynamic} demonstrates that the same recipe for constructing a sheaf as presented here (with different sensors: passive RF sensors and optical cameras) can be used to track moving vehicles under tree cover.  Their resulting sheaf-based tracker smoothly handles sensor hand-off and maintains track custody even when the target is occluded.

2) Can you confirm that the homology of your simplicial complex will be always  Betti_0 = 1, Betti_i = 0 for i >= 1 ? Or can you clarify this aspect? It might suggest further investigations.

Actually, no, the resulting abstract simplicial complex can have an arbitrary Betti number distribution. The following text has been introduced in Section 4, lines 301-312.

            As a general matter, the ASC implies a topological space representing all of these multi-way interactions. Each $d$-dimensional face is a $d$-dimensional hyper-tetrahedron, which are then ``glued'' together or ``attached'' according to the configuration of the sensor interactions. The ASC shown in this example is rather simple, consisting of the ``topmost'' interactions as a single 2-face (the triangle of the ``human'' measurements) and a 1-face (the $RG$ edge for the bear), and then the additional seven sub-faces. This simple structure has a single connected component, and no ``open loops''. But depending on the number of observables informed by a particular sensor, and their configuration, this structure can become arbitrarily complicated, with high order faces and complex connections including open loops or voids also of high dimension. The sheaf theoretical approach can represent all these interactions automatically. While such a ``homological analysis'' of the base space of our sheaf will not be the subject of this paper, it is great interest in computational topology generally \cite{JoCAkS20,Robinson_sheafcanon,EdHHaJ00,GhR07}.

3) I suggest you to add in the conclusion a detailed list of other potential use cases that would benefit of your approach. 

Thank you, the following text has been inserted in the conclusion, lines 760-764.

Since our approach is quite general, it could be applied to many other use cases.  Perhaps the most obvious application is that of locating emergency beacons from downed aircraft \cite{Robinson_sheafcanon} or other hidden radio transmitters \cite{robinson2019hunting}.  More broadly, the idea of consistency radius can be valuable in combining disparate biochemical networks \cite{20161002_ACMBCB}, analyzing the convergence of graphical models and numerical differential equation solvers \cite{Robinson_multimodel,Robinson_qgtopo}, and estimating network flows \cite{nguemo2017sheaf, ghrist2013topological, ghrist2011network}.

Reviewer 3 Report

The paper described a sheaf theoretical approach and its application in details. However, there are two concerns here. 1) the draft is too long.. (Potential) readers might not be interested at most parts of the work; 2) It is unclear what are the novelties of the work. The draft is more like a report, or a case study, the scientific or technology development parts are weak.

Suggest To Reject. 

Author Response

The paper described a sheaf theoretical approach and its application in details. However, there are two concerns here. 1) the draft is too long.. (Potential) readers might not be interested at most parts of the work; 2) It is unclear what are the novelties of the work. The draft is more like a report, or a case study, the scientific or technology development parts are weak.

Thank you for your thoughtful review of our work. We disagree. We find it impossible to address these comments.
